# Network-based machine learning approach to predict immunotherapy response in cancer patients

JungHo Kong[1,4], Doyeon Ha[1,4], Juhun Lee[1,4], Inhae Kim[2], Minhyuk Park [1], Sin-Hyeog Im [1,2,3], Kunyoo Shin [1,3] & Sanguk Kim [1,3 ✉]

Immune checkpoint inhibitors (ICIs) have substantially improved the survival of cancer patients over the past several years. However, only a minority of patients respond to ICI treatment (~30% in solid tumors), and current ICI-response-associated biomarkers often fail to predict the ICI treatment response. Here, we present a machine learning (ML) framework that leverages network-based analyses to identify ICI treatment biomarkers (NetBio) that can make robust predictions. We curate more than 700 ICI-treated patient samples with clinical outcomes and transcriptomic data, and observe that NetBio-based predictions accurately predict ICI treatment responses in three different cancer types—melanoma, gastric cancer, and bladder cancer. Moreover, the NetBio-based prediction is superior to predictions based on other conventional ICI treatment biomarkers, such as ICI targets or tumor microenvironment-associated markers. This work presents a network-based method to effectively select immunotherapy-response-associated biomarkers that can make robust ML-based predictions for precision oncology.

[1] Department of Life Sciences, Pohang University of Science and Technology, Pohang 37673, Korea. [2] ImmunoBiome Inc., Pohang 37666, Korea. [3] Institute of Convergence Science, Yonsei University, Seoul 03722, Korea. [4]These authors contributed equally: JungHo Kong, Doyeon Ha, Juhun Lee. ✉email: sukim@postech.ac.kr

Over the past several years, immune checkpoint inhibitors (ICIs) have drastically improved the clinical treatment of cancer patients[1]. In clinical trials, using ICIs generally induced fewer side effects than chemotherapy with longer-lasting treatment benefits. Accordingly, the use of ICIs has expanded to a constantly growing list of cancer types, including melanoma, bladder cancer, and gastro-esophageal cancer[1]. However, despite the clinical benefits gained from ICI treatments, one major limitation is that only a minority of patients respond to immunotherapy (~30% in solid tumors), and toxicity may occur after ICI treatment[2]. Therefore, a method is needed to identify biomarkers that can detect immunotherapy responders before drug administration, providing information about the clinical use of ICIs and improving the survival of cancer patients[2,3].

A major challenge of precision medicine using immunotherapy is identifying markers from immunotherapy-treated patients that can robustly predict drug responses across multiple cancer patient cohorts. For example, programmed cell death 1 (PD1)/programmed cell death-ligand 1 (PD-L1) expression by immunohistochemistry is a Food and Drug Administration (FDA)-approved companion diagnostic test for various cancer types[4]. Accordingly, many studies have reported a positive correlation between PD-L1 expression and the ICI response in non-small cell lung cancer[5–7]. Strikingly, however, other studies have reported no significant correlation between PD-L1 expression and the ICI treatment response[3,8–10], and some studies have even revealed that ICI responders display low PD-L1 expression levels[3,11]. These inconsistent predictions of previously identified biomarkers necessitate identifying new biomarkers that robustly predict the immunotherapy response. Litchfield et al. recently found that conventional biomarkers can explain only ~60% of the ICI response, suggesting that novel factors are yet to be discovered[12]. Because of the challenges associated with identifying robust biomarkers from immunotherapy-treated patients, many recent studies have focused on identifying biomarkers from cancer patients who were not treated with ICIs, a strategy that benefits from the availability of many samples[13–17]. Despite the success of this approach, a major limitation of these unsupervised learning methods is that markers specific to immunotherapy treatment may not be identified from non-immunotherapy-treated patients, limiting the potential improvements of ICI-based personalized medicine. Therefore, successful methods must be developed to identify biomarkers from ICI-treated patients[3] (e.g., supervised learning methods) and ultimately maximize the benefit of ICI treatment.

Network biology offers a powerful means to identify robust biomarkers. Network-based approach exploits observations that genes with similar phenotypic roles tend to co-localize in a specific region of a protein-protein interaction (PPI) network[18,19]. This tendency has been leveraged to identify gene modules that are much more robust in predicting phenotypic outcomes than using single gene-based approaches[20]. For example, Hofree et al. showed that patients with somatic mutations in similar network regions displayed similar clinical outcomes, although many clinically identical patients share no more than a single mutation[21]. Furthermore, Guney et al. demonstrated that a drug's efficacy can be inferred from the proximity between drug targets and disease genes[22]. In addition, we have previously reported that drug-response biomarkers that predict the overall survival in cancer patients can be identified via network proximity using the pharmacogenomics data of patient-derived organoid models[23]. Altogether, evidences indicate that the network-based approach provides predictive and less noisy biomarkers, but the usefulness of the approach has not yet been validated to predict responses to ICI treatment in a large sample of cancer patients.

Here, we report a network-based machine-learning framework that can (i) make robust predictions across ICI datasets and (ii) identify potential biomarkers. Specifically, we could robustly predict responders and non-responders using the expression levels of network-based biomarkers in more than 700 patient samples, covering melanoma, metastatic gastric and bladder cancer patients treated with ICIs targeting the PD1/PD-L1 axis. To identify robust drug-response biomarkers, we implemented a network-based approach, in which we identified biological pathways located proximal to immunotherapy targets in a PPI network. To measure the generalizability of our biomarkers, we extensively tested within-study cross-validations, as well as across-study predictions. We found that the NetBio-based predictions were more accurate than predictions based on the expression levels of ICI targets including PD1, PD-L1, or cytotoxic T-lymphocyte antigen 4 (CTLA4) and markers associated with the tumor microenvironment, including CD8 T cell, T-cell exhaustion, cancer-associated fibroblast (CAF), and tumor-associated macrophage (TAM) markers. Furthermore, using our network-based transcriptome biomarkers and the tumor mutational burden (TMB), a well-established marker of the ICI response, improved the prediction of the overall survival in ICI-treated bladder cancer patients compared with TMB-based predictions. These findings suggest that network-guided transcriptomic biomarkers can help improve genomic-based ICI response predictions. In summary, our method provides an approach to unveil biomarkers from ICI-treated patients, helping previously identified biomarkers to improve the prediction of the ICI response.

## Results

**Overview of network-based immunotherapy response predictions.** Our previous work supported that biomarkers associated with the anti-cancer drug response are located proximal to the drug targets in a PPI network[23]. Briefly, we found that biomarkers that are associated with a therapeutic effect can be identified from patient-derived organoid models, which were predictive of the drug response in 5-Fluorouracil-treated colorectal cancer and cisplatin-treated bladder cancer patients. Building from our previous work, we aimed to identify biological pathways that are associated with the ICI response by selecting pathways proximal to ICI targets (Fig. 1a, b; Methods). We used the STRING PPI network (STRING score >700)[24], comprising 16,957 nodes and 420,381 edges. First, we applied network propagation, using ICI targets (e.g., PD1 for nivolumab or PD-L1 for atezolizumab) as seed genes, to spread the influence of ICI targets over the network (Fig. 1a and Supplementary Data S1–3). A characteristic of network propagation is that influence scores are higher for nodes closer to ICI targets[25]. Next, we selected genes with high-influence scores (top 200 genes), and identified biological pathways (Reactome pathways[26]) enriched with the genes (Fig. 1b and Supplementary Data S4). We then used the selected biological pathways to predict the immunotherapy response and considered these pathways as Network-Based Biomarkers (NetBio).

To conduct ML-based immunotherapy-response predictions, we used NetBio as input features; as a negative control, we used gene-based biomarkers (i.e., immunotherapy target genes), tumor microenvironment-based biomarkers or pathways selected from data-driven ML approaches (Fig. 1c and Supplementary Data S5, 6). Using the expression levels of the input features, we applied logistic regression to train the ML model. To test the predictive performances of the input features, we measured the performance in predicting (i) the drug response measured by a reduced tumor size after immunotherapy treatment or (ii) the patient's survival. To train an ML model using supervised learning, we used different combinations of training and test datasets to extensively measure

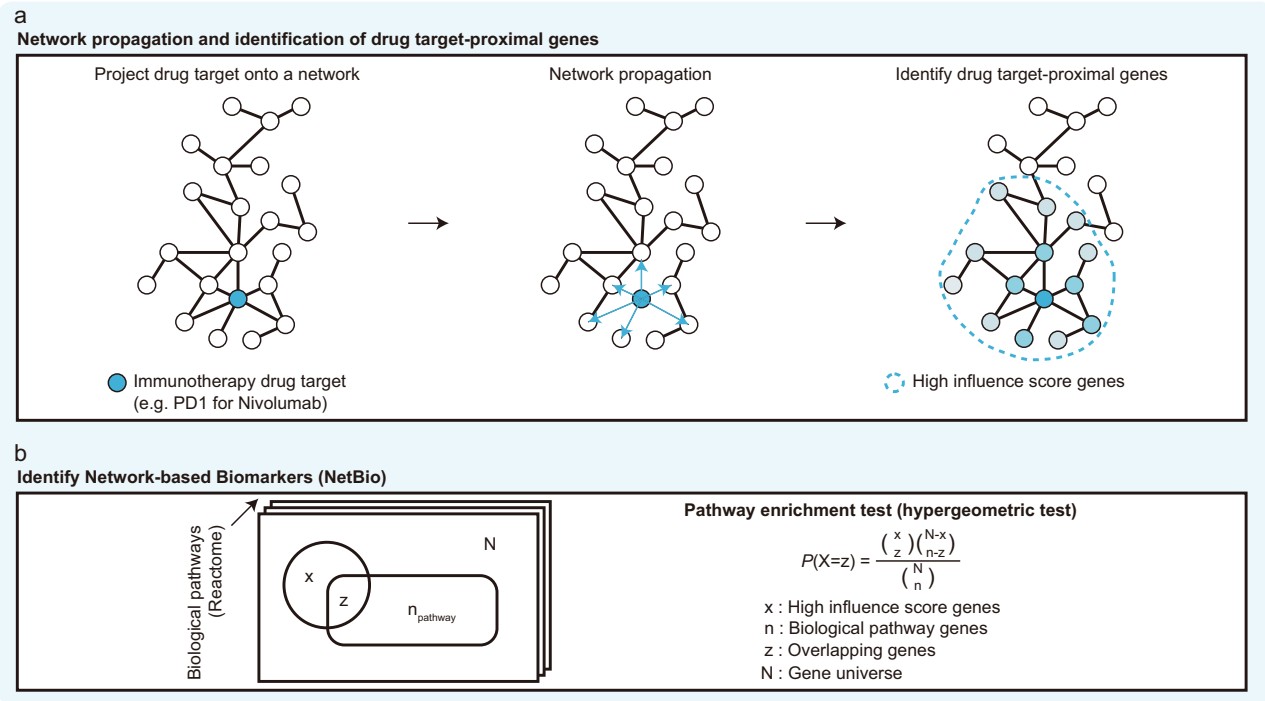

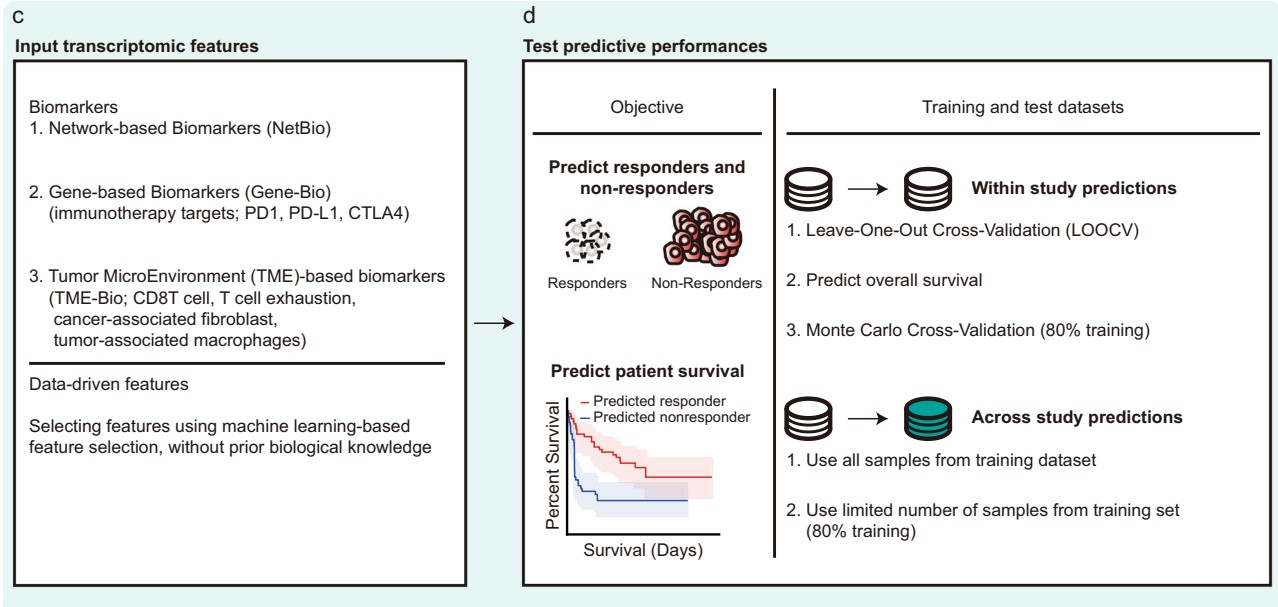

**Fig. 1 A network-based machine-learning (ML) approach to identify immunotherapy-associated biomarkers. a** Network visualization to identify genes proximal to immunotherapy targets in a protein-protein interaction (PPI) network. Immunotherapy targets (e.g., PD-1 for nivolumab) are displayed in blue and projected onto a PPI network, followed by network propagation using drug targets as seed genes. Network propagation is depicted as blue arrows. After propagation, drug target-proximal genes were selected by choosing nodes with high propagation scores (high-influence scores). **b** Identifying Network-based biomarkers (NetBio). Biological pathways (Reactome) enriched with high-influence score genes were selected via the hypergeometric test. **c** Input features used for machine learning to predict immunotherapy responders and non-responders. **d** Overview to measure predictive performances. For prediction objectives, we conducted predictions of the drug response and overall survival. For the training and test datasets, we conducted within-study predictions and across-study predictions.

the consistency of the prediction performances. Specifically, we performed (i) within-study predictions, in which training and test datasets were generated from a single cohort or (ii) across-study predictions, in which two independent datasets were used as training and test datasets (Fig. 1d). Furthermore, we alternated using large or small numbers of training samples to measure the consistency of the prediction performances under various training conditions.

**Within-study cross-validations reveal that NetBio-based ML can make consistent predictions of the ICI treatment response and overall survival.** The transcriptome of our NetBio could make consistent predictive performances to predict the ICI response (Fig. 2). In comparison, we observed less stronger prediction performances when using the expression of drug targets (i.e., PD-1 for nivolumab and pembrolizumab, PD-L1 for

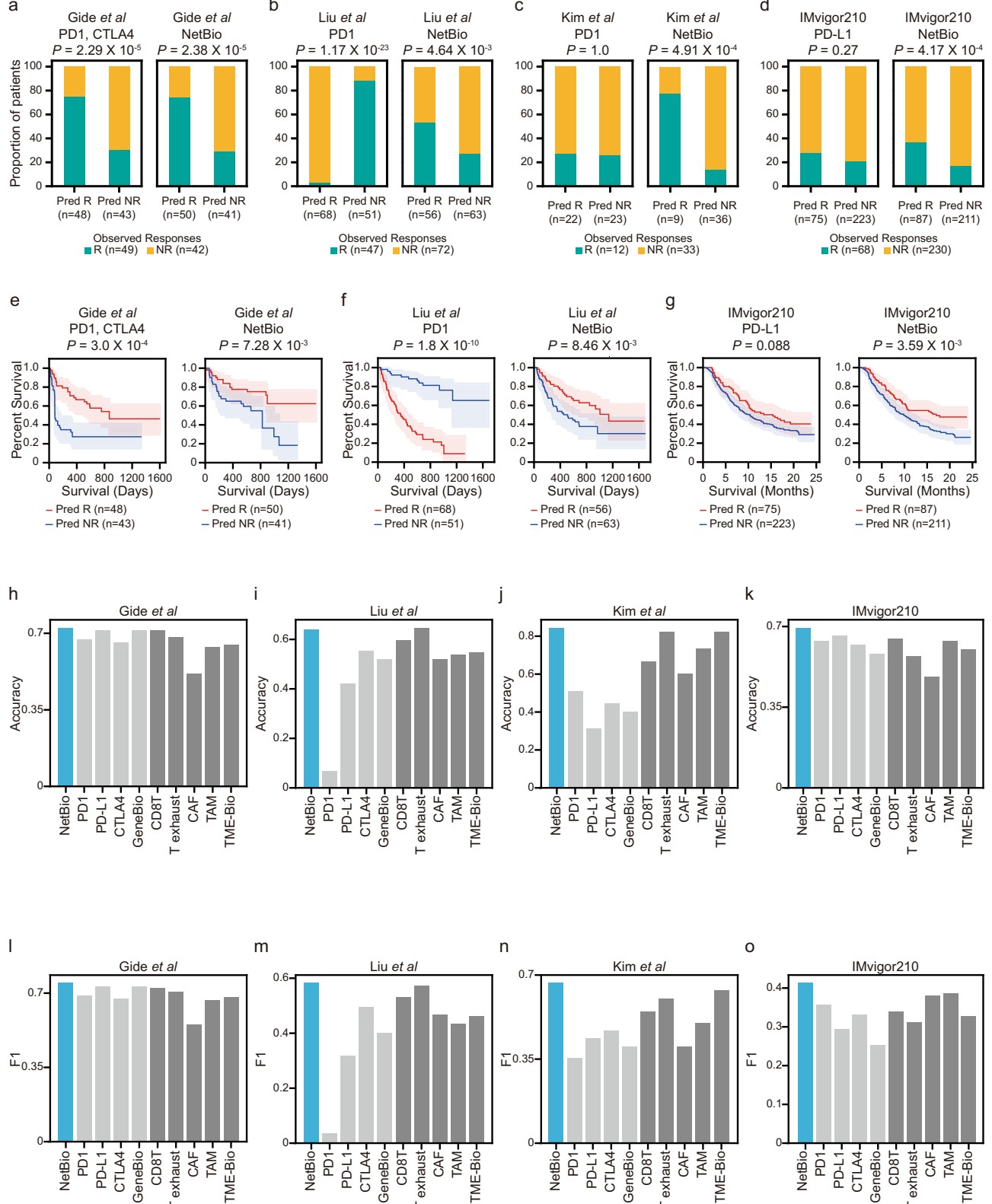

atezolizumab and CTLA4 for ipilimumab-treated patients). We first conducted a leave-one-out cross-validation (LOOCV) to measure the performance using NetBio or other known immunotherapy-related biomarkers (including drug targets). To this end, we used four immunotherapy cohorts—two melanoma cohorts (Gide et al.[27], Liu et al.[28]), one metastatic gastric cancer cohort (Kim et al.[29]) and one bladder cancer cohort

(IMvigor210[30]). The ML model trained using our NetBio consistently made accurate predictions in all four datasets (Fig. 2a–d; Fisher's exact test, $P < 0.05$ was considered significant). By contrast, predictions made using the expression levels of drug targets were less consistent, where drug targets were accurately predictive only in a melanoma cohort (Gide et al.; Fig. 2a) but not in the other three cancer cohorts (Fig. 2b–d). Notably, predictions using

**Fig. 2 Predictions of drug response and overall survival for immunotherapy-treated patients. a–d** Immunotherapy-response prediction using the expression levels of drug targets (PD-1, PD-L1, or CTLA4) or network-based biomarkers (NetBio). Leave-one-out cross-validation (LOOCV) predictions for the (**a**) Gide, (**b**) Liu, (**c**) Kim, and (**d**) IMvigor210 datasets are plotted. Predicted responders (Pred R) and non-responders (Pred NR) are plotted against observed responders (teal) and non-responders (orange). The two-sided Fisher's exact test was used to compute statistical significance. **e–g** Overall survival of predicted responders and non-responders based on LOOCV. The predicted responders and non-responders are depicted in red and blue, respectively. The log-rank test was used to measure statistical significance. The light-colored areas indicated 95% confidence interval of each percent survival. **h–o** LOOCV performance based on NetBio markers; gene-based markers, including PD-1, PD-L1, and CTLA4; and tumor microenvironment (TME)-based markers, including CD8 T cells, T-cell exhaustion, cancer-associated fibroblasts (CAFs), and tumor-associated macrophages (TAMs). GeneBio and TME-Bio include all of the target genes of each category. To quantify performance, we used (**h–k**) accuracy and (**l–o**) F1 score. Source data are provided as a Source Data file.

the expression level of drug targets were inversely predictive in the Liu dataset (Fig. 2b). Furthermore, a prolonged overall survival was consistently observed for patients predicted as ICI responders using our NetBio-based ML in three datasets with overall survival data available (Gide et al.; Kim et al.; IMvigor210; log-rank test $P < 0.05$ was considered significant); using drug target expression predicted the overall survival in only one dataset (Fig. 2e–g). Similarly, we found that NetBio-based LOOCV was able to accurately predict progression-free survival (PFS) in the Gide and Liu datasets (Supplementary Fig. 1a, b; log-rank test, $P < 0.05$ considered significant). By comparison, drug target-based predictions were less consistent in predicting PFS (Supplementary Fig. 1a, b). In particular, prediction based on PD1 expression in the Liu dataset was inversely predictive of PFS (Supplementary Fig. 1b). We also calculated predictions of drug response, overall survival, and PFS in the Liu dataset based on combined expression profiles of PD1 and CTLA4 (Supplementary Fig. 2). The results showed that the combined PD1 and CTLA4 expression levels were not predictive of immunotherapy response, overall survival, or PFS (Supplementary Fig. 2). Altogether, our data showed that the network-based approach, which expands biomarkers to network neighbors of drug targets, improves predictions based on the expression levels of drug targets.

We next compared the predictive performance of our NetBio with other previously identified ICI-related biomarkers and found that our approach was, in most cases, better across all four cancer datasets (Fig. 2h–o). For single gene-based markers, we considered the expression levels of immunotherapy targets (PD-1, PD-L1, or CTLA4). For tumor microenvironment-associated markers, we considered gene sets associated with CD8 T-cell proportions, T-cell exhaustion, CAFs, and TAMs. We also considered using either all the single gene-based markers (GeneBio) or all the tumor microenvironment-associated markers (TME-Bio) to make predictions. We used accuracy and the F1 score to measure the predictive performances of LOOCV and found that NetBio-based predictions were better in 71 of 72 comparisons (98.6%) than predictions using all other biomarkers.

Furthermore, predictions from NetBio were similar to or better than other biomarkers when using fewer training datasets to train ML models. Specifically, we conducted a Monte-Carlo cross-validation. For 100 different iterations, 80% of the samples were randomly selected and used as a training set and the remaining 20% were used as a test set (Supplementary Fig. 3a). In 70 of 72 comparisons (97.2%), our network-based approach showed significantly better or equal performance compared with all other biomarkers (Supplementary Fig. 3b–j; two-sided Student $t$ test $P < 0.05$ was considered significant).

To determine if NetBio can improve predictive performance compared with markers used in clinical settings, such as immunohistochemistry (IHC)-based markers, we compared IHC-based predictions with NetBio-based predictions for the IMvigor210 dataset, which contains both bulk RNA sequencing data and tumor proportion scores (TPS). Compared with TPS,

NetBio performed better in three different prediction tasks, including LOOCV, Monte-Carlo cross-validation (80% training and 20% testing for 100 independent iterations), and overall survival prediction (Supplementary Fig. 4). Our results provide further evidence that using a network-based approach to identify biomarkers can make robust predictions of the ICI response in cancer patients.

**Across-study predictions using NetBio-based ML can make consistent predictions in additional independent melanoma datasets.** Key aspects of an accurate ML model include the following: (i) its ability to generalize to new datasets and (ii) its consistent performance when few training samples are available. First, we observed that the ML model trained using NetBio could make robust predictions when using independent datasets, whereas the predictive performance was poorer when using other biomarkers (Fig. 3). To test the generalizability of our ML model, we used the melanoma dataset from Gide et al. to train the ML model and tested the predictive performance in three independent melanoma datasets (Auslander et al.[13], Prat et al.[31], and Riaz et al.[32]; Fig. 3a). To compute the performance of our model, we used the prediction probability using a logistic regression model. We selected the area under the curve (AUC) of the receiver operating characteristics curve as a performance metric[13–16]. NetBio-based ML showed AUCs >0.7 in two external datasets (Fig. 3b, c; Auslander AUC = 0.79; Prat AUC = 0.72), and 0.69 in the remaining dataset (Fig. 3d; Riaz). In contrast to NetBio-based ML, predictions using other biomarkers displayed highly varying prediction performances (Fig. 3b–d). For example, PD-1 expression showed fewer optimal performances, with the maximum AUC reaching only 0.66 (Fig. 3b–d). Additionally, although predictions using markers of T-cell exhaustion were highly accurate in the Auslander and Riaz datasets (Fig. 3b, d; AUC > 0.7), the prediction performances were slightly better than random expectation in the Prat dataset (Fig. 3c; AUC = 0.58). Moreover, NetBio-based prediction outperformed predictions based on drug targets or tumor microenvironment markers when area under the precision-recall curve (AUPRC) was used as a performance metric (Supplementary Fig. 5). We also observed that NetBio-based prediction performed better than other methods when three independent training datasets were combined into a single dataset (Supplementary Fig. 6), highlighting the robustness of our network-based approach.

Additionally, we found that NetBio improved predictive performance when the training data and test data were drawn from different cohorts. When we used the Liu data to train the machine-learning model and then tested the predictive performance in three different cohorts (Supplementary Fig. 7a), NetBio-based predictions outperformed predictions based on other ICI-related biomarkers in 88.5% (23/26) of comparisons (Supplementary Fig. 7b–d). These results suggest that regardless of the datasets used to train the machine-learning model, NetBio

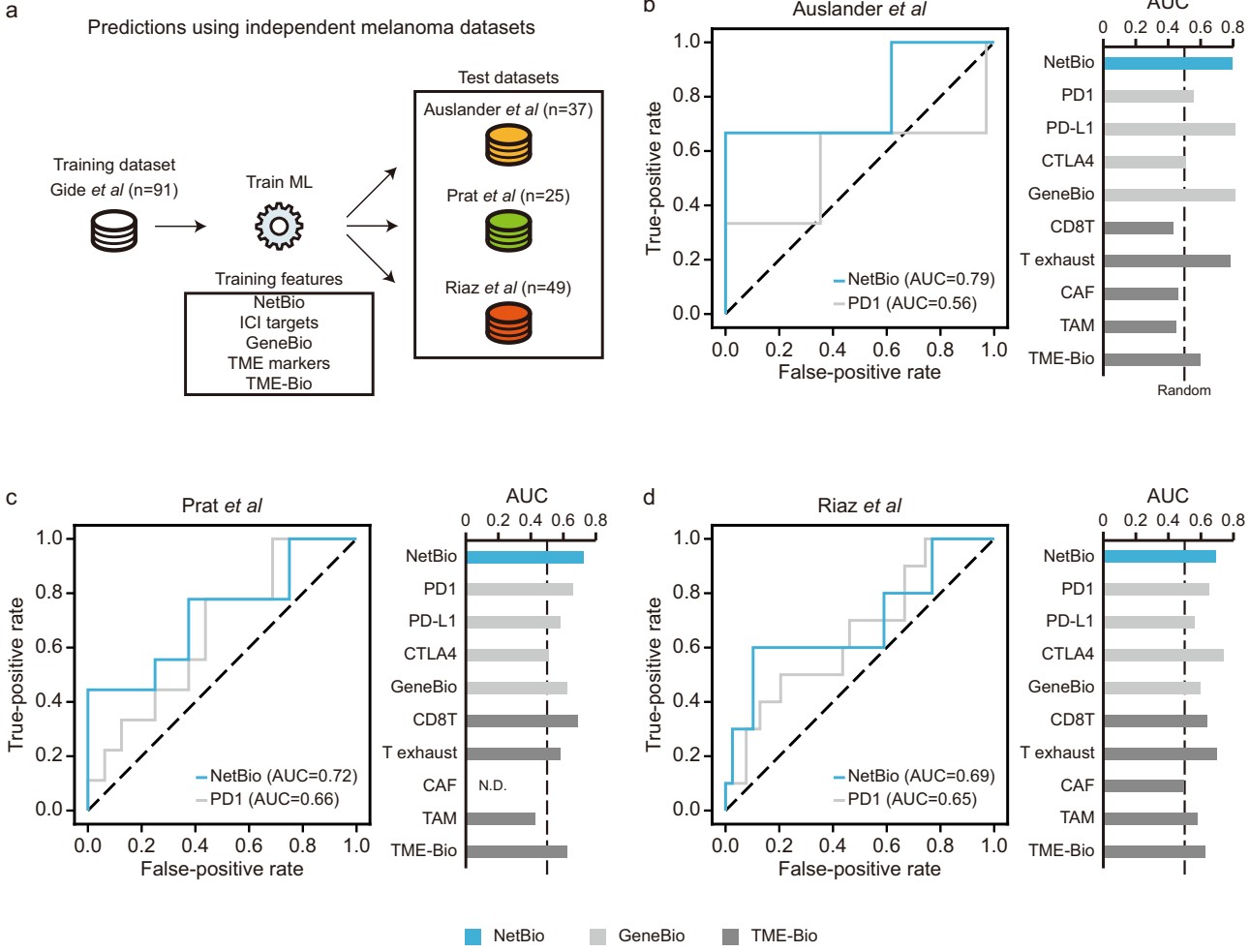

**Fig. 3 Predictive performance in three independent melanoma datasets. a** Overall scheme of immunotherapy-response predictions in three independent datasets. The datasets and transcriptomic features used to train and test the machine-learning models and the number of samples for each dataset are displayed. **b–d** The area under the receiver operating characteristic curve (AUC) for the (**b**) Auslander, (**c**) Prat, and (**d**) Riaz datasets is shown. The random expectation, equaling an AUC of 0.5, is displayed as dotted lines. Expression profiles of cancer-associated fibroblast (CAF) marker genes were not available in the Prat dataset. N.D. not detected. Source data are provided as a Source Data file.

can improve predictive performance compared with drug target-based or tumor microenvironment-based biomarkers.

Next, we tested the performance of NetBio-based predictions using data on cancer recurrence after anti-PD-1 treatment in a recent cohort of melanoma patients (Huang et al.[33]) (Supplementary Fig. 8a). We found that regardless of the training dataset used (Gide or Liu), NetBio-based markers accurately predicted cancer recurrence after ICI treatment (Supplementary Fig. 8b, c; Gide to Huang AUC = 0.78, Liu to Huang AUC = 0.8). These results suggest that NetBio-based machine-learning can be a useful framework for predicting ICI responses in new datasets.

Next, we tested whether the ML model can make robust predictions even when fewer training samples are available. Again, NetBio-based ML with smaller sample sizes made consistent predictions compared with GeneBio or TME-Bio-based ML models. To test this, for 100 iterations, we randomly sampled 80% of patients from the training dataset (Gide dataset) to train the ML model and tested the prediction performance in three external melanoma datasets (Supplementary Fig. 9a). Our biomarkers showed statistically significantly better or equal performance in 49 of 54 comparisons (Supplementary Fig. 9; 90.7%). Only PD-L1 expression in the Auslander dataset, CTLA4 in the Riaz dataset, and CD8 T-cell exhaustion markers in the

Riaz datasets displayed prediction performances that were better than NetBio-based predictions when using AUC as the measure of performance, but these biomarkers (PD-L1, CTLA4, and CD8 T exhaustion markers) were inconsistent in their predictions in the other melanoma datasets (Supplementary Fig. 9d–i).

**NetBio-based predictions outperform other state-of-the-art methods of drug response prediction.** Next, we compared NetBio-based prediction with other state-of-the-art methods for immunotherapy-response prediction[13,14,16,17] as well as a deep neural network (DNN)-based method[34] (see the Methods). We first tested the predictive performance for LOOCV. We found that NetBio-based prediction was better than the other methods in 33 of 34 comparisons (Supplementary Fig. 10; 97.1%). For across-study predictive performance, NetBio-based prediction was better than the other methods in 17 of 18 comparisons (Supplementary Fig. 11; 94.4%). These results suggest that NetBio can improve prediction of ICI treatment response compared with other biomarkers.

**NetBio-based predictions outperform purely data-driven feature selection approach.** A major limitation of using data-driven ML models for clinical applications is its inability to consistently

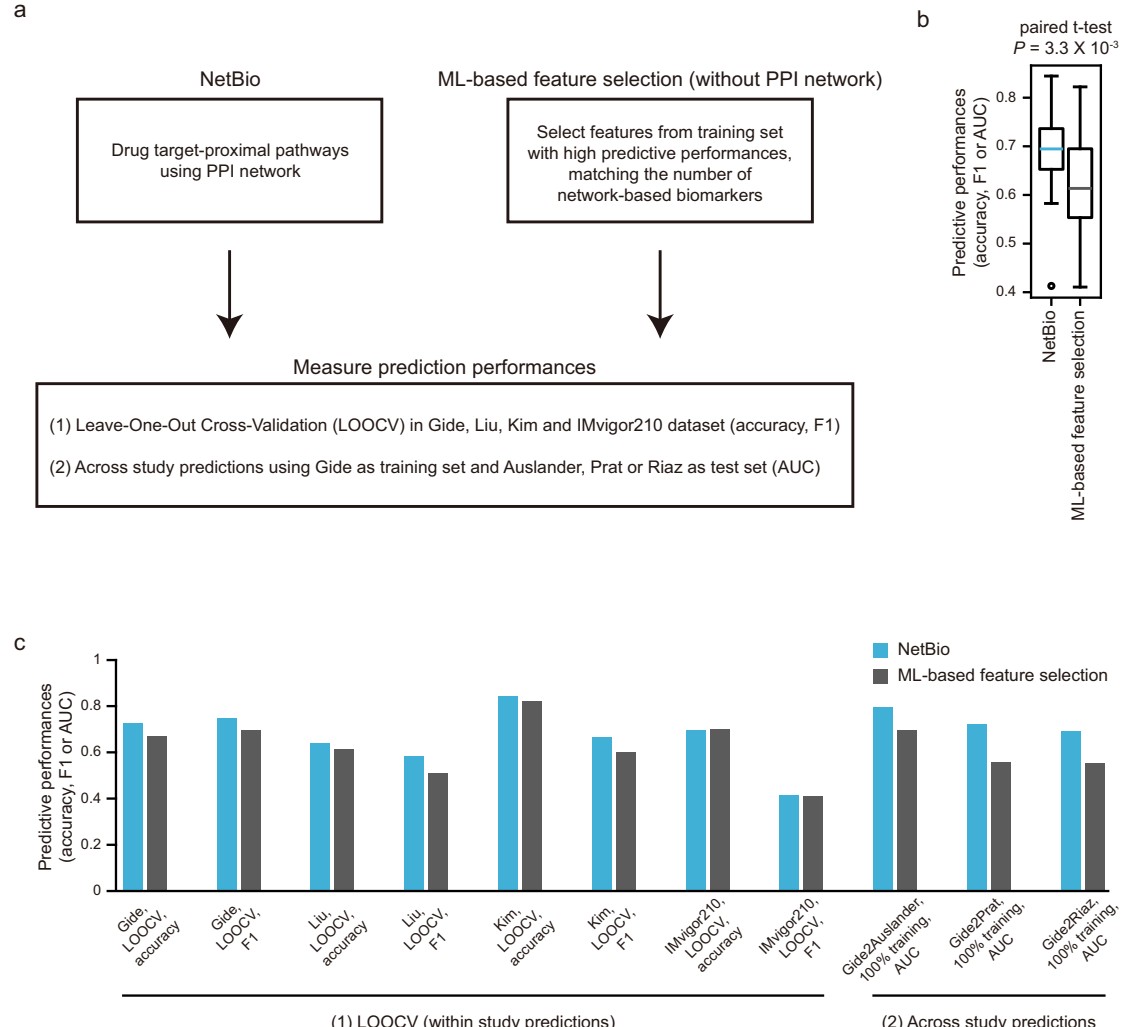

**Fig. 4 Comparison of predictive performance using machine learning-based feature selection. a** Overall scheme for comparisons. **b** Overall predictive performance using NetBio-based or machine learning-based feature selection for 11 independent tests. Statistical significance was measured using the two-sided paired-sample *t* test. Boxplot shows median value, interquartile range (IQR) as bounds of the box and whiskers that extends from the box to upper/lower quartile ± IQR × 1.5. **c** Bar plots of predictive performances in 11 different tests, using accuracy, F1 score, or AUC as a metric to quantify performance. Source data are provided as a Source Data file.

perform in new datasets, despite performing well in training datasets. Thus, we tested whether the addition of prior biological knowledge, representing a PPI network in this study, can improve feature selection compared with purely data-driven feature selection approaches. The NetBio-based ML model enables consistently improved prediction performances compared with purely data-driven ML predictions (Fig. 4). In detail, for the data-driven ML model, we selected K number features (where K equals the number of NetBio) that best distinguish responders and non-responders in a training dataset and used the selected features to train the ML model (Fig. 4a; Methods). In 11 different tasks, we found that NetBio-based predictions showed significantly better performance than features from ML-based feature selection (Fig. 4b; two-sided paired Student *t* test $P = 3.3 \times 10^{-3}$). Furthermore, performance improvements were consistently observed when predicting across melanoma cohorts (across-study predictions; Fig. 4c), suggesting that network-guided selection can help reduce the overfitting of ML models. This observation suggests that network-guided feature selection can provide robust features compared with those from purely data-driven feature selection. Altogether, our result suggests that robust transcriptomic

biomarkers can be identified by leveraging network-based biomarker selection.

**NetBio-based predictions recapitulate the immune microenvironment in external The Cancer Genome Atlas (TCGA) datasets.** Because NetBio robustly performed the best across distinct cohorts encompassing three different cancer types, we investigated whether NetBio-based predictions can recapitulate the immune microenvironment that is associated with immunotherapy responses. We tested how NetBio-based predictions were correlated with immune contextures in the TCGA datasets[35] (Fig. 5a). Specifically, we used the Gide or Liu dataset (melanoma cohorts) to predict ICI responses in melanoma patients in the TCGA dataset (TCGA SKCM), Kim dataset (gastric cancer cohort) to predict TCGA gastric cancer (TCGA STAD), and IMvigor210 dataset (bladder cancer cohort) to predict TCGA bladder cancer (TCGA BLCA) patients and correlated the predicted drug response with (i) the tumor mutation burden (TMB) or (ii) immune contextures of TCGA patients (Fig. 5a). For immune contextures, we used immunogenic scores computed by Thorsson et al.[36]. The entire correlation results for NetBio-based

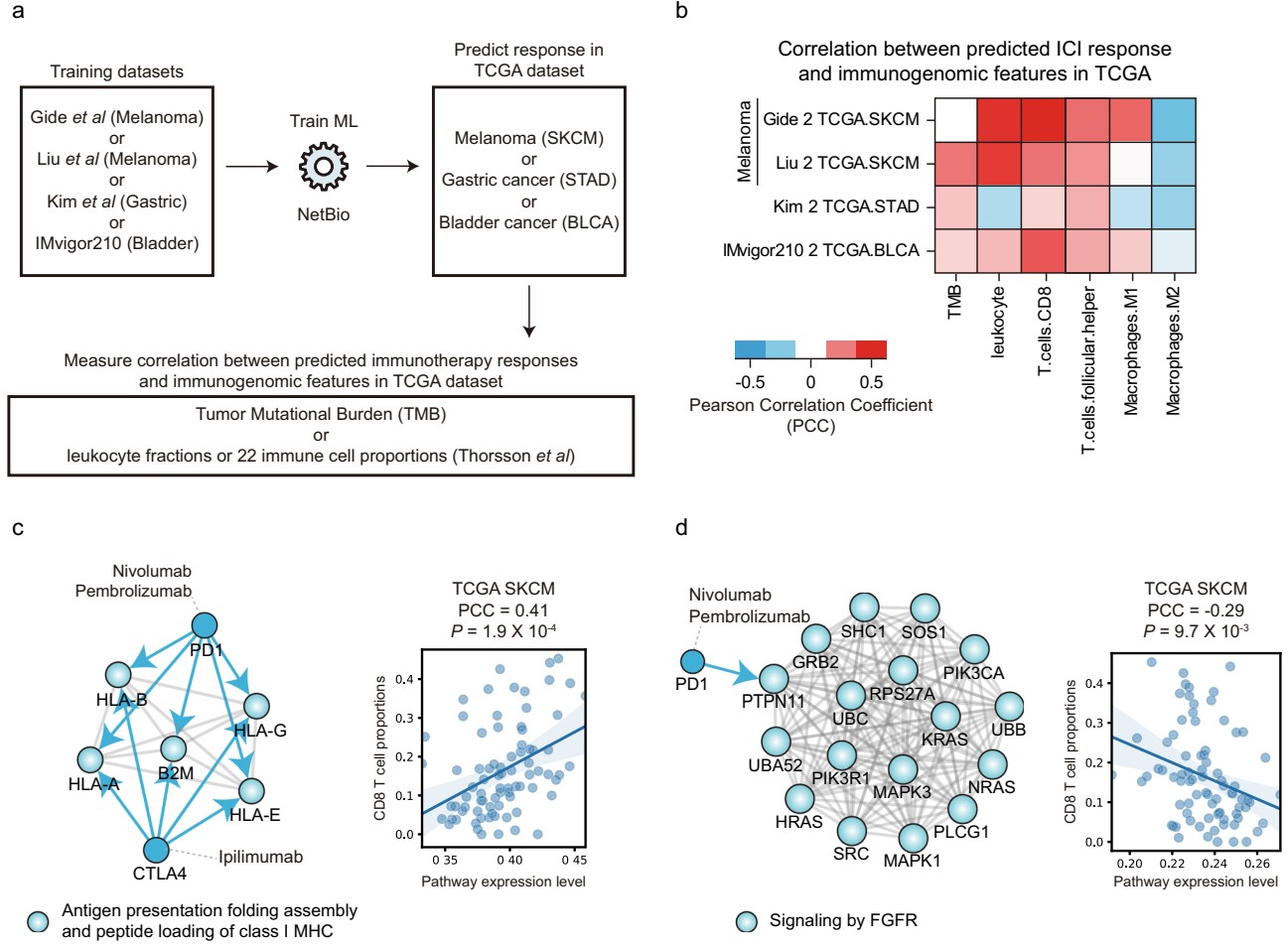

**Fig. 5 NetBio-based predictions recapitulate the immune microenvironment. a** Research scheme to compute the correlation between NetBio-based predictions and immunogenic features in the TCGA dataset. **b** Correlation between the predicted drug response using NetBio and immunogenic features in the TCGA cohort. Correlation was measured using Pearson's correlation coefficient (PCC). **c**, **d** NetBio pathways identified from (**b**) Gide et al. and (**c**) Liu et al. are shown. The scatterplot displays the correlation between pathway expression and immunogenic features. The light-colored areas indicated 95% confidence interval of linear regression line. The PCC and correlation *P* values are shown. Source data are provided as a Source Data file.

predictions versus TMB or immune contextures are available in Supplementary Fig. 12.

NetBio-based predictions successfully recapitulated the immune microenvironments (Fig. 5b). We speculated that the correlation results from Gide and Liu cohorts have common characteristics because they both concern melanoma patients. As expected, they exhibited similar immune microenvironment characteristics, including a high positive correlation with leukocyte fractions and CD8 T-cell proportions, and a high negative correlation with M2 macrophage proportions (Fig. 5b). By contrast, we observed reduced correlations with immune signatures when we merged three TCGA cancer types into a single cohort for analysis (Supplementary Fig. 13), suggesting the importance of considering cancer-type specificity. Moreover, we also found that regardless of the training dataset used (Gide or Liu), patients with the "immune" phenotype in the SKCM TCGA dataset[37] were likely to be predicted ICI responders based on NetBio markers (Supplementary Fig. 14), suggesting that predicted ICI responders have high immune infiltration levels. Interestingly, the correlation between predictions based on the two different training sets was weak (Supplementary Fig. 15), suggesting that (i) ICI responders may have distinct immune cell infiltration mechanisms and (ii) multiple molecular subtypes may exist within melanoma patients.

We further investigated which NetBio pathway was responsible for the high correlation with immune cell proportions. The pathway features of greatest importance from ML training (top 10 greatest feature importance with positive coefficient) using the Gide dataset (Supplementary Fig. 16) revealed that "antigen presentation folding assembly and peptide loading of class I MHC" displayed the highest positive correlation with CD8 T-cell proportions (Fig. 5c and Supplementary Fig. 16; PCC = 0.41). This finding was expected because antigen presentation by antigen-presenting cells or tumor cells induces the infiltration of CD8 T cells. When using the Liu dataset, among pathways of greatest importance (top 10 greatest feature importance with negative coefficient), "FGFR signaling" showed the highest correlation with CD8 T-cell proportions (Supplementary Fig. 17), where the expression level of the pathway was negatively correlated with the cell proportions (Fig. 5d and Supplementary Fig. 17; PCC = −0.29). Moreover, we found that the expression level of "FGFR signaling" was lowest in SKCM TCGA patients with the immune subtype (Supplementary Fig. 18), suggesting that low expression of FGFR signaling is associated with high immune infiltration. Consistent with our findings, recent studies have suggested that fibroblast growth factor 2 depletion can lead to increased T-cell recruitment, enabling tumor regression[38]. Our results here suggest the following: (i) non-identical CD8 T-cell

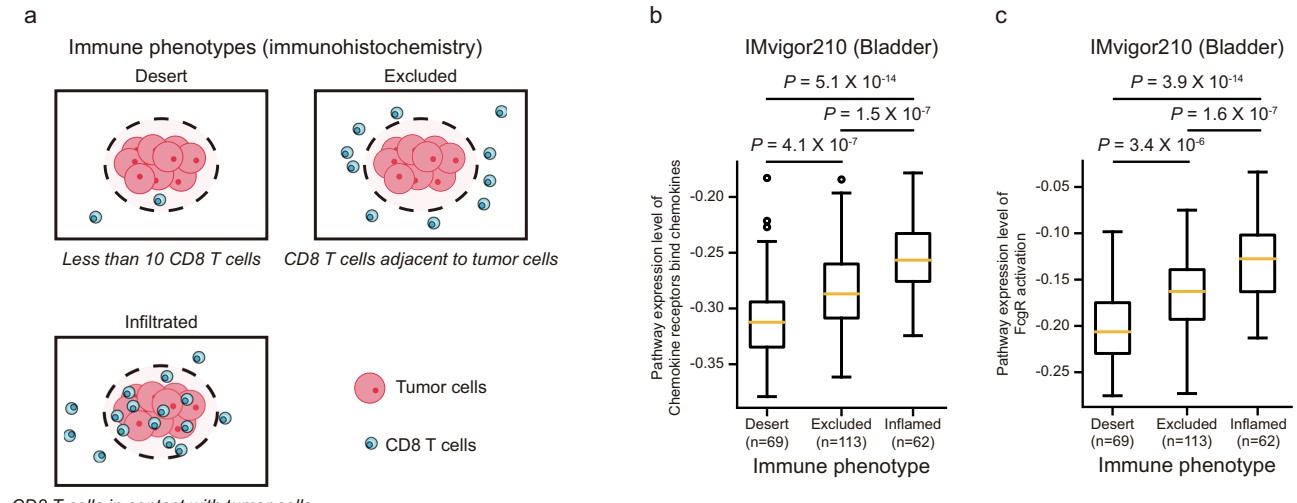

**Fig. 6 The expression levels of NetBio pathways are consistent with immunohistochemistry-based immune phenotypes in bladder cancer.**
**a** Categorization of immune phenotypes using immunohistochemistry. **b**, **c** Expression levels of NetBio pathways in various immune phenotypes. For NetBio pathways, chemokine receptors bind chemokines (**b**) and FcgR activation (**c**) are shown. The two-sided Mann–Whitney U test was used to compute statistical significance for differential pathway expression levels across different immune phenotype patient groups. Boxplot shows median value, interquartile range (IQR) as bounds of the box and whiskers that extends from the box to upper/lower quartile ± IQR × 1.5. Source data are provided as a Source Data file.

recruitment mechanisms may exist in melanoma and (ii) NetBio can robustly capture CD8 T-cell recruitment in tumor samples, even when different melanoma cancer cohorts are used to train an ML model.

NetBio pathways were also identified that were consistent with the immune microenvironment in gastric and bladder cancer. In gastric cancer, NetBio-based predictions were highly correlated with follicular helper T-cell proportions (Fig. 5b). Among pathways of greatest importance from the Kim cohort, a high expression level of "mitotic G2-G2-M phases" was associated with high follicular helper T-cell proportions (Supplementary Figs. 16, 19). Consistent with our results, a previous study reported that the differentiation of helper T cells was regulated by the cell cycle pathway[39]. In bladder cancer, we found that NetBio-based predictions were positively correlated with the leukocyte fractions (Fig. 5b). Accordingly, the NetBio pathways demonstrated chemotaxis (i.e., chemokine receptors bind chemokines) and phagocytosis (i.e., FcgR activation), which are functions closely associated with immune infiltration (Supplementary Figs. 16, S20). These pathways displayed a high correlation with leukocyte fractions in TCGA bladder cancer patients (Supplementary Fig. 20a, b; PCC > 0.6). Our results suggest that the immune microenvironments can be captured using NetBio pathways in gastric cancer and bladder cancer.

**Expression levels of NetBio pathways are associated with immune cell infiltration in bladder cancer patients.** Because infiltration of immune cells was reported to be closely associated with anti-cancer drug responses in bladder cancer[30,40], we asked whether expression levels of NetBio pathways in the bladder cancer TCGA dataset (Supplementary Fig. 20) are associated with immune cell infiltration levels. In bladder cancer patients, we validated that both chemotaxis and phagocytosis pathways (i.e., chemokine receptors bind chemokines and FcgR activation, respectively) are associated with immune infiltration in the PD-L1 treated bladder cancer cohort, using additional IHC-based results (Fig. 6). We used immune phenotypes in the IMvigor210 dataset[30]. Specifically, we used distinct immune phenotypes including (i) immune desert

(fewer than 10 CD8 T cells), (ii) excluded (CD8 T cells adjacent to tumor cells), and (iii) infiltrated (CD8 T cells in contact with tumor cells) phenotypes[30] (Fig. 6a) and compared the expression levels of chemotaxis and phagocytosis pathways with the immune phenotypes (Fig. 6b, c). The immune infiltrated phenotype displayed the highest expression level of the pathways compared with the immune desert or excluded phenotypes (Fig. 6b, c; Mann–Whitney U $P < 0.05$), suggesting that the NetBio pathways can capture leukocyte infiltration fractions in bladder cancer. Altogether, our results suggest that NetBio can consistently unveil pathways related to the immunotherapy response-associated immune microenvironment.

**Combining NetBio expression levels with the tumor mutation burden (TMB) in an ML model improves the prediction of PD-L1 inhibitor-treated bladder cancer patients.** Although a high TMB level is associated with increased benefits of ICI treatment, ICI responders and non-responders often show significant overlap of TMB levels, suggesting that TMB alone is not a sufficient predictor of the ICI response[4,41,42]. Thus, we tested whether combining our NetBio with TMB-based predictors improves prediction performance (Fig. 7a). Combining the NetBio expression levels and TMB improved the prediction of the overall survival in bladder cancer patients treated with atezolizumab, which is a PD-L1 inhibitor (Fig. 7b, c and Supplementary Fig. 21). Using LOOCV to predict the ICI treatment response with only the TMB to train the ML model, the 1-year percent survival difference between the predicted responder group and predicted non-responder group was 18% (Fig. 7b; log-rank test $P = 2.0 \times 10^{-3}$; the 1-year percent survival rates for the predicted responder and predicted non-responder group was 60.8% and 42.8%, respectively). The 1-year percent survival difference was increased to 22.3% when using both the TMB and NetBio (Fig. 7c; the 1-year percent survival rates for the predicted responder and predicted non-responder group were 64.4% and 42.1%, respectively), as well as improvements in log-rank test statistics ($P = 2.02 \times 10^{-4}$).

Next, we observed that the combined predictors correctly reclassified non-responders from predicted responders using

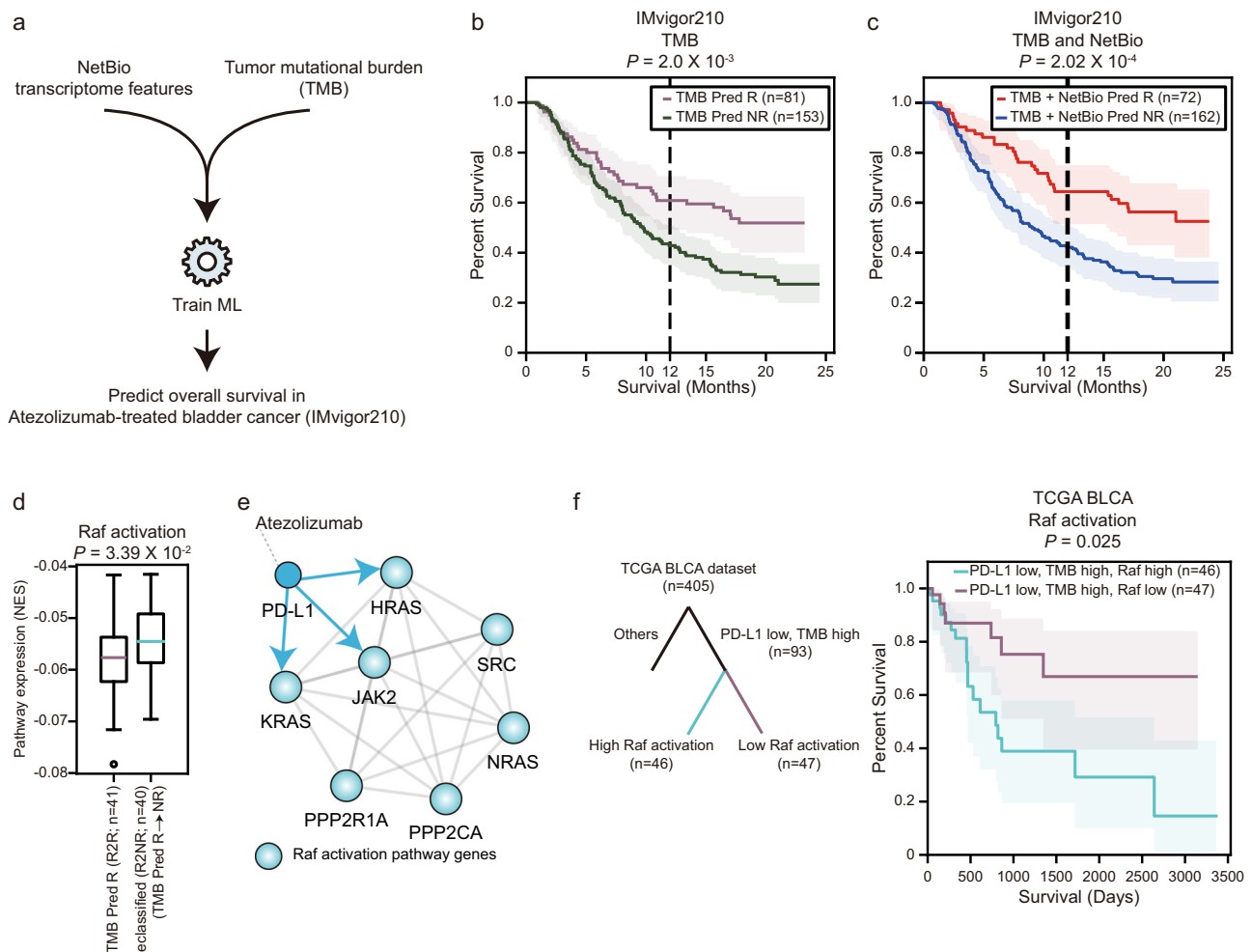

**Fig. 7 Combining network-based transcriptome features and the tumor mutation burden (TMB) improves the prediction of the overall survival in PD-L1 inhibitor (atezolizumab)-treated bladder cancer patients. a** Overall scheme of the network-based transcriptome and TMB combined predictions. **b**, **c** Prediction of the overall survival using (**b**) TMB-only and (**c**) a combined ML model. Statistical significance was measured using the log-rank test. The light-colored areas indicated 95% confidence interval of each percent survival. **d** Differential expression of the Raf activation pathway between predicted responders from TMB-only predictions and the reclassified subgroup from combined ML model predictions. Boxplot shows median value, interquartile range (IQR) as bounds of the box and whiskers that extends from the box to upper/lower quartile ± IQR × 1.5. The two-sided Student's t test was used for statistical significance. **e** Network representation of the atezolizumab target (PD-L1) and Raf activation pathway. **f** Association between PD-L1 expression, the TMB levels and the expression levels of the Raf activation pathway with overall survival in bladder cancer patients (TCGA BLCA dataset). The light-colored areas indicated 95% confidence interval of each percent survival. Source data are provided as a Source Data file.

TMB alone (R2NR; Supplementary Fig. 22) and correctly reclassified responders from predicted non-responders from TMB-alone predictions (NR2R; Supplementary Fig. 22). R2NR patients exhibited a lower overall survival than the predicted responder group when using only the TMB (Supplementary Fig. 22b); the 1-year percent survival decreased to 51.2% (log-rank test $P$ value = 0.07). Similarly, the 1-year percent survival increased to 57.1% in NR2R patients and displayed a statistically significant increase in the overall survival compared with the predicted non-responders using TMB-based predictions (Supplementary Fig. 22c; log-rank test $P = 1.94 \times 10^{-2}$). Altogether, our results suggest that TMB combined with NetBio transcriptomic features can improve the correct classification of responders and non-responders.

Having observed improved prediction performances, we sought to identify a feature responsible for the improvements in the prediction performance. We first observed that the TMB levels remained similar in the reclassified subgroups (Supplementary Fig. 23), suggesting that the TMB levels are not a

confounding factor in the improved prediction of the overall survival. To identify a transcriptomic feature associated with resistance to immunotherapy in the high TMB group, we investigated differentially expressed pathways between predicted responders using TMB-based predictions (i.e., high TMB group) and the R2NR group. The Raf activation pathway was significantly differentially expressed between the two subgroups (Fig. 7d; two-sided Student's t test $P = 3.39 \times 10^{-2}$). In detail, patients who were predicted as non-responders from the combined prediction model (i.e., R2NR patients) displayed higher expression of Raf activation pathway components. From the PPI network, components of the Raf activation pathway, including HRAS, KRAS, and JAK2, were direct neighbors of PD-L1 (Fig. 7e), suggesting that this pathway may exert a mechanistic effect during drug treatment.

To further examine the potential usefulness of the Raf activation pathway as an ICI-treatment biomarker, we analyzed the association among PD-L1 expression, the TMB and the expression level of Raf activation components with the overall

survival in an external TCGA bladder cancer dataset ($n = 405$). Specifically, we tested whether Raf activation affected overall survival when (i) the PD-L1 expression was low, simulating PD-L1 inhibition, and (ii) the TMB level was high. The Raf activation pathway had a statistically significant impact on the overall survival in bladder cancer patients exhibiting low PD-L1 expression and high TMB levels (Fig. 7f; $P = 0.025$). Importantly, higher expression of the Raf activation pathway was associated with poor overall survival, a finding that is consistent with PD-L1 inhibitor-treated patients exhibiting resistance to the treatment (Fig. 7d, f). Altogether, our results suggest that (i) network-based transcriptomic biomarkers can help improve TMB-based immunotherapy-response predictions and (ii) ICI response biomarkers can be identified using network-based approaches.

## Discussion

In this study, we tested whether the network-based biomarker discovery pipeline can make robust predictions of immunotherapy treatment. NetBio-based ML demonstrated consistent predictive performance, whereas GeneBio, TME-Bio-based predictions, or features identified from purely data-driven approaches, showed less optimal performances (Figs. 2–4). Our work is further supported by previous studies utilizing PPI networks to (i) increase the detection of robust biomarkers and (ii) improve the prediction of clinical outcomes in cancer patients. For example, Leiserson et al. used network modules to identify cancer-type-specific and pan-cancer driver genes[43]. Additionally, Cheng et al. recently reported that disease-associated germline mutations that alter protein-protein interactions are highly correlated with cancer patient survival and the response to anti-cancer drugs[44], a finding that is similar to our previous observation that disease-associated variants are frequently located at protein interaction interfaces[45]. Furthermore, we have previously demonstrated the usefulness of the PPI network to understand gene-phenotype relationships[46–53], including the identification of oral disease-[46] and mitochondrial disorder[47,50]-associated variants. Taken together, our findings offer a network-based ML model that robustly predicts the immunotherapy response in cancer patients.

Because a complete and accurate map of the PPI network is critical for network-based approaches[19], we asked how the predictive performance would be affected if a smaller network (STRING score >900) were used to identify NetBio pathways. We compared the NetBio pathways found using STRING > 900 (NetBio 900) to those found using STRING > 700 (NetBio 700) and observed high overlap coefficient scores across four cohorts (Gide, Liu, Kim, and IMvigor210) (Supplementary Fig. 24). These results show that the majority of the pathways in NetBio 900 were included in NetBio 700, suggesting that the pathways are conserved. Moreover, we found that although NetBio 900 had reduced predictive performance compared with NetBio 700, the network-based approach with the smaller network was still effective in predicting ICI response (Supplementary Figs. 25, 26). In a within-study LOOCV task, the predictive performance of NetBio 900 was equal to or better than that of other ICI biomarkers, such as GeneBio and TME-Bio, in 32 of 36 comparisons (Supplementary Fig. 25; 88.9%). Furthermore, in across-study predictions, NetBio 900 performed better than other ICI biomarkers in 40 of 54 comparisons (74.1%) (Supplementary Fig. 26). These results suggest that although the performance of ICI response prediction declines when a smaller network is used, the network-based approach still performs better than target gene-based and tumor microenvironment-based biomarkers. Also, the reduced predictive performance resulting from the use of an incomplete network highlights the importance of network coverage for identifying drug-response biomarkers. Additionally, continuous development

of network propagation algorithms will help improve tasks of precision medicine since the algorithms have been successfully applied to identify disease genes and drug target[54] s. In this study, a random walk with restart was employed. However, various algorithms of network propagation have been recently proposed to account for degree bias of protein interaction networks[55,56]. These methods have a potential to find diseases modules with improved performance of identifying disease genes, drug target candidates, and biomarkers for drug response.

We also identified that NetBio-based predictions can consistently recapitulate immune microenvironments that are associated with the immunotherapy response. Across three different cancer types (melanoma, gastric cancer, and bladder cancer), we found that NetBio-based predictions were consistently positively correlated with the proportions of anti-tumor leukocytes such as CD8 T-cell proportions, whereas the proportions of pro-tumor leukocytes, such as M2 macrophages, were consistently negatively correlated with NetBio-based predictions (Fig. 5b). Our prediction results are consistent with previous study findings because (i) ICI treatment aims to reinvigorate CD8 T cells such that higher CD8 T-cell proportions lead to increased ICI treatment efficacy[30,57]; (ii) M2 macrophages suppress CD8 T cells such that higher proportions of M2 macrophages result in the resistance to ICI treatment[58]. Furthermore, NetBio-based predictions consistently recovered CD8T cell proportions even when different melanoma cohorts (Gide et al. or Liu et al.) were used to train the ML model (Fig. 5b). Altogether, our results suggest that NetBio pathways, which are network neighbors of ICI targets, robustly capture patients' immune composition from transcriptome data. Given the consistency of our results, a future research opportunity would be to apply the network-based approach with higher-resolution sequencing techniques (e.g., single-cell RNA sequencing) that enable consideration of important aspects of the immune microenvironment, including immune cell proportions or cell states[59].

One might ask whether combining multiple cancer types in a comprehensive dataset might improve the performance of NetBio-based prediction. We found that combining all cancer types into a single comprehensive dataset did not improve the performance of ICI response prediction, suggesting the importance of cancer type-specific ICI response mechanisms. First, we tested whether gene expression patterns of network-based binding partners to the ICI drug targets were similar across cancer types (see the Methods). We found that transcriptome similarity was high between two melanoma cohorts (median transcriptome similarity of 0.39 and 0.41 for ICI responders and non-responders, respectively), whereas it was lower between cohorts with different cancer types (Supplementary Fig. 27). We next used ComBat[60] to remove batch effects among four independent datasets (Gide, Liu, Kim, IMvigor210) and combined the datasets for NetBio prediction. We found that the LOOCV performance of the combined NetBio markers was decreased compared with that of NetBio markers based on each individual dataset (Supplementary Fig. 28). These results suggest that expression-based biomarkers of ICI treatment response differ across cancer types.

Although the identification of drug-response biomarkers has traditionally focused on genomic markers[17], we tested whether NetBio-based transcriptomic features, when combined with genomic features, can improve the prediction of immunotherapy responses. Specifically, we selected the TMB for genomic feature because a higher mutation burden is likely to increase neoantigen presentation, which can subsequently increase T-cell infiltration and ICI treatment efficacy[4]. Combining the TMB levels with NetBio-based transcriptomic features improved the prediction of the overall survival in PD-L1 inhibitor-treated bladder cancer patients (Fig. 7b, c; Supplementary Fig. 22). Consistent with our predictions in bladder cancer, we observed that combining

NetBio and TMB levels improved the prediction of overall survival in a melanoma cohort (Supplementary Fig. 29). Our results suggest that combining various omics datasets can improve the prediction of the response to ICI treatment in cancer patients. Additionally, combining TMB with NetBio provided transcriptomic biomarkers responsible for improved ICI-response prediction in bladder cancer. We identified the "Raf activation" pathway, which is a downstream pathway of the Epithelial Growth Factor Receptor (EGFR) gene, as a transcriptomic feature in the IMvigor210 cohort (Fig. 7d–f). In detail, up-regulation of the pathway was correlated with a poor response to ICI treatment (Fig. 7d). Similar to our findings, multiple clinical trials have reported that lung cancer patients harboring activating EGFR mutations show resistance to PD-1 and PD-L1 inhibitor treatments[61]. Because the Raf signaling pathway is a direct downstream pathway of EGFR, activation of the Raf pathway may also be responsible for the poor response to ICI treatments. Further studies on the role of the Raf activation pathway in the immunotherapy response in bladder cancer will be required to confirm this possibility.

We envision that our work here opens up interesting new research opportunities for precision medicine using ICI treatment. For example, we have developed an ML method that trains directly from ICI-treated samples (i.e., supervised learning), whereas most state-of-the art techniques use ML models that learn from non-ICI-treated samples to predict the response to ICI treatment (i.e., unsupervised learning)[13–17]. Because supervised and unsupervised learning uses different cancer patients to train ML models, both learning approaches may complement each other, leading to improved prediction performances when used together (e.g., the semi-supervised approach). As a proof of concept, combining NetBio-based predictions with those from the unsupervised learning approach by Lee et al.[15] using gene-gene synthetic lethal interactions can improve the prediction of the ICI response (Supplementary Fig. 30). Specifically, we found that the performance of combined predictions was improved across all tested conditions when predictions from supervised learning (NetBio) and unsupervised learning (Lee et al.) showed low correlation with each other (Supplementary Fig. 30b), suggesting that both learning methods can learn distinct, yet ICI-treatment-relevant, biological signals. Since biological outcomes of immunotherapy are highly complex, a method relying on a single omics feature has a limitation in predicting patient response to immunotherapy treatments. Combining a network-based machine-learning model with diverse omics layers would make better clinical results. As more sequencing data of tumor samples become available for both ICI-treated and non-ICI-treated cancer patients, we hope that our work here, along with other previous and future ML methods, can facilitate major improvements in precision oncology.

## Methods

### Curation and pre-processing of patient data
We collected the data of the following eight different patient cohorts treated with ICIs targeting the PD-1/PD-L1 axis: (i) Gide et al. (nivolumab-, pembrolizumab-, and/or ipilimumab-treated melanoma; $n = 91$)[27]; (ii) Liu et al. (nivolumab- or pembrolizumab-treated melanoma; $n = 121$)[28], (iii) Kim et al. (pembrolizumab-treated metastatic gastric cancer; $n = 45$)[29]; (iv) IMvigor210 (atezolizumab-treated bladder cancer, $n = 348$)[30]; (v) Auslander et al. (anti-PD-1- and/or anti-CTLA4-treated melanoma; $n = 37$)[13]; (vi) Prat et al. (nivolumab- or pembrolizumab-treated melanoma; $n = 25$)[31]; (vii) Riaz et al. (nivolumab-treated melanoma; $n = 49$)[32]; (viii) Huang et al. (pembrolizumab-treated melanoma; $n = 13$)[33]. For the Prat et al. dataset, we only considered melanoma samples. For the Riaz et al. dataset, we only used expression samples collected before drug treatment. For the Huang dataset, we considered patients without recurrence to be ICI responders and patients with recurrence to be ICI non-responders. Detailed information on the drug-response labels used in the study is available in Supplementary Table 1. The datasets were not combined into a single comprehensive dataset unless noted. We did not generate any new data for this study, so no additional ethics approval was required.

Regarding the TCGA dataset, we used the following: (i) TCGA SKCM (melanoma; $n = 103$); (ii) TCGA STAD (stomach adenocarcinoma; $n = 375$); and (iii) TCGA BLCA (bladder cancer; $n = 405$). Gene expression data (HTSeq—Counts), somatic mutation data, and clinical data (i.e., overall survival data) were downloaded using the TCGAbiolinks R package[62]. To calculate the TMB in TCGA cancer patients, we used the following equation from Wang et al.[63]:

$$\text{TMB}_{\text{patient}} = T_{\text{patient}} 2.0 + NT_{\text{patient}} \times 1.0 \quad (1)$$

where $T_{\text{patient}}$ is total number of truncating mutations and $NT_{\text{patient}}$ is the total number of non-truncating mutations. For truncating mutations, we considered nonsense mutations, frame-shift deletion or insertion and splice-site mutations. For non-truncating mutations, we used missense mutations, in-frame deletion or insertion, and nonstop mutations.

For the pre-processing of gene expression data, we calculated the gene expression levels using read counts from the IMvigor210, Auslander, Prat, Riaz, and TCGA datasets, which were normalized using trimmed means of M-values normalization[64] from the edgeR[65] R package. For other datasets, we used normalized expression values provided by Lee et al. (https://zenodo.org/record/4661265)[15]. To estimate the pathway expression levels, we used Reactome pathways downloaded from the MSigDB database[26] and performed single-sample GSEA (ssGSEA)[66] using the GSVA R package[67]. We used the normalized enrichment score (NES) to estimate the pathway expression levels of each sample (Supplementary Data S7).

To classify samples into responders and non-responders, we used response evaluation criteria in solid tumors (RECIST) criteria, where complete response (CR) and partial response (PR) were classified as responders and stable disease (SD) and progressive disease (PD) were classified as non-responders, as in previous studies[15,34,68–71]. For dataset that did not provide or use RECIST criteria (Auslander dataset), we used responder and non-responder classification from the original paper. The clinical outcome data used in the paper are provided in Supplementary Data S8.

### Preparation of the PPI network
We downloaded the human PPI network from the STRING database v.11.0. (https://string-db.org/)[24]. To leverage high-confidence PPIs, we considered links with interaction scores greater than 700[20,23]. Next, for network-based analysis in this manuscript, we used the largest connected component of the PPI network, resulting in 16,957 nodes and 420,381 edges. The largest connected component was computed using the NetworkX python module[72]. We used Cytoscape (v.3.7.1) for network visualization[73].

### NetBio detection
The detection of NetBio pathways comprises two steps: (i) the detection of ICI target-proximal genes in the PPI network and (ii) detection of biological pathways (Reactome pathway[26]) proximal to ICI targets (i.e., NetBio pathways). First, we identified ICI target-proximal genes via network propagation using the page-rank algorithm from the NetworkX python module[72]. We used one for ICI targets and zero for all other genes in the network as an input for the personalization parameter in the page-rank algorithm. Default settings were used for any other parameters for the page-rank algorithm (damping factor = 0.85). After network propagation, we considered the top 200 genes with highest influence scores as ICI target-proximal genes.

Next, we detected biological pathways located proximal to ICI targets using ICI target-proximal genes. We computed the gene set enrichment test that specifically calculates how many ICI target-proximal genes are included in each pathway. We used the hypergeometric test to obtain statistical significance. Finally, we selected pathways significantly enriched with ICI target-proximal genes using an adjusted P value of <0.01. The Holm-Sidak test was used for multiple hypothesis testing. We computed hypergeometric test statistics and the adjusted P value using scipy[74] and statsmodels[75] python modules, respectively. The number of NetBio pathways selected for the Gide, Liu, Kim, and IMvigor210 cohorts was 472, 323, 292, and 353, respectively. The NetBio pathways are provided in the Source Data. We used the expression profile of all NetBio pathways to train a logistic regression classifier.

To test whether ICI response is dependent on network connectivity, we tested if the connectivity of the binding partners of ICI drug targets (PD1, PD-L1, and CTLA4) was correlated with ICI efficacy. To measure ICI efficacy for each binding partner, we used each patient's binding partner expression level and ICI response and computed the AUC of gene expression and ICI response to define ICI efficacy. We observed that in four different ICI-treated cohorts, ICI efficacy did not correlate with the connectivity of the binding partners (Supplementary Fig. 31; P value < 0.05 considered significant). These results suggest that a gene's degree centrality is not a confounding factor when predicting ICI response.

Furthermore, we found that expression profiles of NetBio pathways did not significantly change from prior to treatment to during treatment (Supplementary Fig. 32a, b). Using a melanoma cohort (Riaz et al.), we identified differentially expressed pathways (DEPs) by comparing pre-treatment and during-treatment expression profiles (Supplementary Fig. 32a). We found that compared to all pathways available from the Reactome database, DEPs were not enriched in NetBio pathways (Supplementary Fig. 32b; two-sided Fisher's exact test P = 0.5). This suggests that the expression levels of NetBio pathways do not necessarily change during ICI treatment.

**Measuring the performances of ML predictions**. Throughout the manuscript, we used logistic regression to train ML models, implemented in Scikit-learn in Python[76]. Specifically, we used the l2 regularized logistic regression (LR) model. We also tested the predictive performances of NetBio-based machine-learning using Support Vector Classifier (SVC), random forest (RF), and deep neural network (DNN) models. We found that SVC and RF models performed similarly to the LR-based model, whereas the LR-based model was more generalizable to new datasets than DNN-based models (Supplementary Fig. 33, Supplementary Fig. 34). To train ML models, we used the expression levels of genes/pathways against drug responses (classified as responders and non-responders). To select optimal hyperparameters for LR-based model, we conducted fivefold cross-validation in a training dataset by iterating the regularization parameter (C) from 0.1 to 1 in 0.1 intervals. We used "balanced" parameters for class weight hyperparameters to reduce class imbalance effects. To identify optimal hyper-parameters, we used the GridSearchCV function from the Scikit-learn module[76]. The optimal hyperparameters identified during LOOCV are provided in the Source data. The gene/pathway expression levels are z-score-standardized before ML training/testing to minimize the batch effect between cohorts, where z-score standardization was done for each gene/pathway across samples of the same cohort[23,77]. For across-study predictions, the distributions of predicted responses are provided in Supplementary Fig. 35. Z-score-standardized expression data were used to combine three training datasets for across-study predictions (Supplementary Fig. 6).

For LOOCV, we considered cohorts that agree with the following criteria: (i) cohorts with more than 30 samples and (ii) at least 10 samples for both responders and non-responders. Four datasets remained after applying the criteria (Gide et al., Liu et al., Kim et al., and IMvigor210). We used the LeaveOneOut function from the Scikit-learn module to split the training and test datasets[76]. The accuracy, precision, F1, true-positive rate, true-negative rate, false-positive rate, false-negative rate, sensitivity, and specificity of LOOCV are given in the Supplementary Tables (Supplementary Tables 2–5).

For predictions based on genes (GeneBio) and the tumor microenvironment (TME-Bio), we used gene expression levels to train/test the ML model. For GeneBio, we used the expression levels of PD-1, PD-L1 or CTLA4. For TME-Bio, we used the gene expression levels of markers of (i) CD8 T cells[78], (ii) T-cell exhaustion[14], (iii) CAFs[79], and (iv) TAMs (M2 macrophages)[14]. The detailed gene list for each marker and references for the gene lists are provided in the Source Data.

To test the performance of data-driven ML predictions, we conducted feature selection using the SelectKBest function from Scikit-learn[76] ("f_classif" was used for the score function parameter). We selected K number of reactome pathways, where K equals the number of NetBio pathways. To train and test the data-driven ML model, we used the pathway expression levels. Notably, SelectKBest function-based feature selection was conducted using the training dataset.

To further investigate the association between DEPs and drug responses, we tested whether DEPs could accurately predict responders and non-responders in various melanoma datasets (Supplementary Fig. 32a). We used the expression profiles of DEPs to train a machine-learning model and conducted (i) within-study prediction (LOOCV) and (ii) across-study prediction (Supplementary Fig. 32c, h). We observed that in some cases, DEPs provided information to differentiate responders from non-responders (Supplementary Fig. 32c–p); however, in most cases, NetBio-based predictions were better than DEP-based predictions (Supplementary Fig. 32c–p). These results suggest that the baseline gene expression profiles associated with drug response may not necessarily change after ICI treatment.

**Comparison with other state-of-the-art methods**. We used EASIERscores[16] provided by the original authors. We computed IMPRES scores[13] using pairwise comparisons of 15 gene pairs, as was done in the original manuscript[13]. The TIDE scores[14] were computed using the TIDEpy python package (https://github.com/liulab-dfci/TIDEpy). For TMEsubtypes scores[17], we used the microenvironment subtypes of melanoma patients, which are provided in the original publication[17]. We used an l2 regularized logistic regression model to test the performance of the four state-of-the-art prediction methods[13,14,16,17]. For the DNN-based method[34], 10 sets of hyperparameters were selected at random from the hyperparameter grid and fivefold cross-validation was conducted to select the best-performing hyper-parameters. The hyperparameter grid used in our work is provided in Supplementary Table 6. For the activation function, we used the hyperbolic tangent (tanh) for all hidden layers except the final output layer, where we used the sigmoid function.

**Comparing NetBio pathway expression with IHC phenotypes in the bladder cancer dataset (IMvigor210)**. We analyzed the IMvigor210 dataset,[30] which contains both gene expression profiles and IHC staining data. The immune phenotypes based on IHC staining were (i) immune desert, (ii) excluded, and (iii) infiltrated. The immune phenotypes were determined based on the pre-valence of CD8 T cells and infiltration patterns with respect to malignant epi-thelial cells[30]. The presence of CD8 T cells was detected using an anti-CD8 antibody (rabbit monoclonal clone SP16)[30]. The expression levels of

"Chemokine receptors bind chemokines" and "FcgR activation" were used based on ssGSEA NES values.

**Combining TMB levels and NetBio to predict overall survival**. We used TMB levels and expression levels of NetBio pathways to predict ICI response. Both TMB levels and expression levels of NetBio pathways were z-score standardized prior to machine-learning training (l2 regularized logistic regression). For the IMvigor210 dataset (Fig. 7), we used the mutation burden per megabase as the TMB level. For the Liu dataset (Supplementary Fig. 29), the number of nonsynonymous mutations was used as the TMB level.

**Calculating the expression similarity of network-based binding partners to the ICI drug targets**. We used the expression levels of the network neighbors of ICI targets (PD1, PD-L1, and CTLA4) to measure transcriptome similarity between cohorts (Supplementary Fig. 27). We defined the transcriptome similarity as follows: (1) for each patient, we computed Spearman rank correlation to all patients in another cohort; (2) we took the maximum value from the Spearman correlations; (3) we iterated steps (1) and (2) for all patients in both cohorts.

**Calculating prediction performances for the combined model using NetBio-based predictions and predictions from the synthetic lethal relationship (SELECT)**. The SELECT score[15] was provided by the original authors. SELECT uses synthetic lethal and synthetic rescue relationships between two genes identi-fied from non-ICI-treated cancer samples. Before combining the SELECT score with NetBio-based predictions (using the prediction probability from LOOCV), we first computed Spearman's correlation between the two prediction scores. In the Kim et al. cohort (metastatic gastric cancer), the two prediction scores showed no correlation with each other (Spearman's correlation rho = 0.28; P = 0.16; Supple-mentary Fig. 30b), suggesting that the two different prediction models captured distinct biological signals.

To combine the SELECT score with NetBio-based predictions (Supplementary Fig. 30a), we used the linear weighted model by Zhang et al.[80]:

$$\text{Combined score} = w(\text{NetBio predictions}) + (1 - w)(\text{SELECT score}) \quad (2)$$

where w is the linear weight ranging from 0 to 1 in 0.1 intervals (Supplementary Fig. 30b). We used the AUC of the receiver operating characteristics curve as a performance metric.

**Statistical analysis and software**. Fisher's exact test, Mann–Whitney U test, and two-sided Student t test were used for data analysis and generation of P values. Log-rank test was used to compute statistical differences in overall survival and progression-free survival. For correlation analysis, we used Pearson correlation unless otherwise noted. All analyses were done in python 3.6.12. Python packages used are pandas (1.1.15), numpy (1.19.2), scipy (1.5.4), matplotlib (3.3.3), sklearn (0.24.2), lifelines (0.25.7), networkx (2.5), statsmodels (0.12.2), and pytorch (1.7.l + cu110).

**Reporting summary**. Further information on research design is available in the Nature Research Reporting Summary linked to this article.

## Data availability

For the Gide et al.[27], Huang et al.[33], Kim et al.[29], and Liu et al.[28] datasets, we used normalized expression values and drug responses provided by Lee et al.[15]. The data sets are available without requesting the access in Zenodo [https://zenodo.org/record/4661265]. Access of original data could be obtained from PRJEB23709, GSE123728, PRJEB25780, and the Supplementary Data 2 of Liu et al.[28]. The IMvigor210 dataset was downloaded from the original paper [http://research-pub.gene.com/IMvigor210CoreBiologies][30]. The Auslander dataset[13] and Riaz dataset[32] were downloaded from the GEO repository[81] under the accession numbers GSE115821 and GSE91061, respectively. The Prat dataset[31] was downloaded from the Supplementary Material of the original paper (Table S1). The TCGA datasets were downloaded using the TCGAbiolinks R package [https://bioconductor.org/packages/release/bioc/html/TCGAbiolinks.html][62]. The human PPI network was downloaded from the STRING database v.11.0. [https://string-db.org/][24]. The Reactome pathways[26] were downloaded from the MSigDB[66] database. All data used in this study are publically available. Source data are provided with this paper.

## Code availability

The source codes for reproduction of the results were developed in python 3.6.12. and are available at a GitHub repository (https://github.com/SBIlab/NetBio)[82].

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

## Acknowledgements

We thank all of the members of the Kim laboratory for helpful discussions. We also thank Prof. Joo Sang Lee for providing the SELECT scores. Moreover, we are grateful for Professor Federica Eduati for providing the EASIER score. This work was supported by grants to S.K. from the Korean National Research Foundation (2021R1A2B5B01001903 and 2020R1A6A1A03047902), Ministry of Oceans and Fisheries ("Omics based on fishery disease control technology development and industrialization" (20150242)), and IITP (2019-0-01906, Artificial Intelligence Graduate School Program, POSTECH), and to J.K. from POSTECHIAN fellowship.

## Author contributions

J.K., D.H., J.L., I.K., S.I, K.S., and S.K. conceived and designed the experiments. J.K., D.H., J.L., and M.P. curated the patient data. J.K., D.H., J.L., and I.K. performed the experiments. J.K., D.H., J.L., I.K., M.P., S.I, K.S., and S.K. analysed the data. J.K., D.H., J.L., I.K., M.P., S.I, K.S., and S.K. wrote the paper.

## Competing interests

The authors declare no competing interests.
