## [Peer Review File · Nature Communications]

REVIEWER COMMENTS

Reviewer #1 (Remarks to the Author): Expert in network medicine

--In that ~30% of patients respond to immune checkpoint inhibitor therapy, I suggest you change the phrase "only a few patients respond to immunotherapy..." to "the minority of patients respond to immunotherapy...."

--How different were the gene expression patterns for network-based binding partners to the ICI drug targets among the different cancers? Would there be any (at least statistical) value in combining all cancers in one comprehensive data set and repeating the analysis?

--Have you considered developing a strategy for weighting the interaction network edges based at least on the connectivity of the drug target and of each of its binding partners? In the absence of data on protein concentrations and association constants for each binding partner, greater connectivity might be expected to correlate inversely with efficacy if, for example, only one or a few binding partners drive the primary response phenotype, as the other binding partners 'dilute' the effect by competition.

--I suggest that Figure S3 (or at least S3b) be included in a figure in the main text.

--Do you have any transcription profiles from tumors after treatment? If so, could you validate the putative pathways predicted to govern response by analyzing changes in gene expression with treatment?

--Please comment on the potential benefits of using single-cell RNAseq analysis as transcriptomic data sources (as perhaps a next step). Using mean-field values for gene expression, especially in cancers, runs the risk of failing to detect a small population of cells (or of gene products) that primarily drive the response. The individual cell-based transcriptome or a limited population of cells whose transcriptomes are sufficiently similar may offer the opportunity for even better predictive accuracy if the analysis were also coupled with network-based information.

Reviewer #2 (Remarks to the Author): Expert in immunotherapy response and biomarkers in melanoma

The authors have submitted a manuscript analyzing a curated dataset of 700 ICI treated patients with available clinical outcomes and transcriptomic data

Used NetBio predictions in melanoma, gastric cancer, bladder cancer

Compared to “other ICI treatment biomarkers, such as ICI targets”

“Presents a network-based method to effectively select IO-response-associated biomarkers that make robust ML-based predictions for precision oncology”

In the introduction, the statement: “programmed cell death 1 (PD1)/programmed cell death-ligand 1 (PD-L1) expression by immunohistochemistry is a Food and Drug Administration (FDA)-approved companion diagnostic test for various cancer types. REF: 4”

- Unfortunately, melanoma (one of the cancer types selected in this manuscript) is not one of those types. So the utilization of PD-L1 as a prognostic biomarker in melanoma does not make sense and undermines the subsequent analysis.

- References mentioned 4, 6-8 in PD-L1 underperformance are also not in melanoma.

Would agree that better biomarkers needed. Question remains why the comparison of performance is then compared to these clearly underperforming biomarkers (which again in the case of melanoma is not used clinically at all).

Would also agree that network based models have the potential to outperform others, but in doing this comparison combining multiple disparate disease types may be a challenge in data interpretation.

NetBio created by:

- Used STRING PPI network using ICI targets as seed genes
- Selected genes with high influence scores (top 200)
- Used Reactome to identify pathways enriched with the genes
- Used the pathways (not genes) to predict ICI responses

After creating NetBio

- Negative control: gene based biomarkers (IO target genes), TME genes, or pathways from ML approaches
- Compared to: 1: drug response measured by reduced tumor size after ICI tx or 2: OS
 - o Is (1) RECIST or not?
 - o For RECIST data, methods state aggregation: CR/PR and SD/PD – SD patients are often actually considered responders since they are non-progressing
 - o Other datasets just used their R vs NR status (did that agree with CR/PR and SD/PD categories)? Should be the same across all data sets (and often these variable do differ across studies)
 - o Also OS is not a great marker for response to IO (patients will see many other lines of treatment), usually use PFS or even cancer-specific survival
- Training and validation from same data set
- Used 2 melanoma cohorts and 1 bladder cohort to measure performance

Feedback:

Overall, the authors bring together 700 quite heterogeneous datasets including different tumor types (unclear why these types chosen) with different immune checkpoint inhibitor treatments. The analysis moves back and forth from different datasets and it is unclear how data sets were integrated or chosen for analysis (see specific comments on melanoma – Liu et al. included in some places and not in others). These inconsistencies are quite troubling.

Additionally, the efforts to bridge across different tumor types is quite challenging and makes data interpretation difficult. Beyond melanoma, there are certainly other published data sets in each of the tumor types listed for validation. Why not included here?

In this vein, subsequent integration with TCGA to “validate IO responses” is unfounded. SKCM TCGA (and likely others) did not include patients on systemic therapies as these were “landscape biology” efforts. The authors attempt to correlate their signatures/predictions with features in the TCGA, but none of which make any sense. In SKCM TCGA, they correlate with Ag presentation (in one dataset) and FGF (in another). There is no effort to align with the actual TCGA phenotypes, one of which is “immune” suggesting their data may not align with known biology.

After creating NetBio

- Negative control: gene based biomarkers (IO target genes), TME genes, or pathways from ML approaches
- Compared to: 1: drug response measured by reduced tumor size after ICI tx or 2: OS

Is (1) RECIST or not?

For RECIST data, methods state aggregation: CR/PR and SD/PD – SD patients are often actually considered responders since they are non-progressing

Other datasets just used their R vs NR status (did that agree with CR/PR and SD/PD categories)? Should be the same across all data sets (and often these variable do differ across studies)

Also OS is not a great marker for response to IO (patients will see many other lines of treatment), usually use PFS or even cancer-specific survival

In Fig. 6 the authors switch to another TCGA and use IHC to compare “excluded” and “immune included” tumors – an abrupt transition with no real explanation.

Finally, Fig. 7 looks only at bladder cancer and integrates TMB.

Summary:

Overall, despite interesting methodologies and approaches, there is little consistency and synergy of data analysis and validation across the different data sets and tumor types. This undermines the real-world applicability of this type of analysis.

I would recommend the authors narrow to one histology, enlarge the included datasets and align more closely with the known phenotypes in that field. The current all-inclusive approach increases the total “n” of samples, but likely blurs what is likely distinct biology.

Specific comments from manuscript:

Across-study predictions:

- Here used Gide et al to train and Auslander, Prat, and Riaz to test performance
- Why was Liu et al excluded here? It was used previously as a representative melanoma dataset

NetBio based predictions recapitulate immune microenvironment in external TCGA datasets

- Here tested how NetBio-based predictions correlated with immune contextures in TCGA
- Used Gide or Liu dataset to predict ICI response in melanoma patients in TCGA dataset (therapeutic response is NOT present in TCGA dataset)

Went on to look at gastric and bladder cancer.

Finally, the group combined NetBio with TMB in aPD1 treated bladder cancer.

Major issues:

Not all immune checkpoint inhibitors are the same, biomarkers likely different. This is reflected in analysis in the Liu et al paper (patients previously tx with aCTLA4 vs aPD1 naive patients) and different histologies:

Melanoma

-Gide et al. (nivo-, pembro-, and/or lpi)

-Liu et al. (nivo- or pembro- (some previously treated with aCTLA4, subanalysis in this paper)

-Auslander et al (aPD1 and/or anti-CTLA4)

-Prat et al (nivo-, pembro-)

-Riaz et al (nivo-)

Gastric

-Kim et al (pembro-)

Bladder

-IMvigor210 (atezo-)

Major Comments on Figures:

-Fig 2A/B, E/F: Why compare to PD1, CTLA4 in Gide and only PD1 in Liu? Makes no sense.

-Fig 3: why do the authors suddenly exclude the Liu data set and only train with Gide et al? Liu disappears from this analysis altogether

-Fig 4a: Gide now again compared to Liu

-Fig 4c: now Gide vs all (no Liu) – this back and forth makes no sense to me

-TCGA has no response data (these were all untreated patients) – so the correlation of immune signatures with IO response is speculative at best. Also undermined by differences amongst the datasets, even in the same tumor type (e.g. melanoma) – Gide, Ag presentation; Liu, FGFR signaling.

-Also SKCM TCGA has clearly delineated “immune” subtype melanomas, no evidence of analysis or alignment with the published SKCM tumor groups.

-Fig 6: now compared to excluded and inflamed phenotypes using IHC. No consistency in criteria used across these different analyses.

-Fig. 7: suddenly focuses on just bladder cancer aPD-L1 treated patients and combination with TMB. Seems out of the blue for the paper.

Minor:

-Recommend an experimental overview/schema since including and aggregating data from a wide variety of studies (and integrated differently in different sections)

-Text in figures is fuzzy and hard to read

Reviewer #3 (Remarks to the Author): Expert in network medicine and systems biology

Kong and colleagues present a timely method to predict response to several immune checkpoint inhibitors (ICIs) used for melanoma, gastric cancer and bladder cancer. The method relies on building a logistic regression classifier using biological pathways that are enriched with genes that are in close vicinity of the drug targets in the human interactome. The classifier is trained and tested using data from the same cohorts (via leave-one-out cross-validation) as well as using data from independent cohorts. When expression of genes in the biological pathways enriched with genes proximal to drug targets (so called Netbio – network based biomarkers) were used, the authors show modest but clinically significant improvement on the predicted responders (in around 90% of different comparisons conducted across data sets) and survival rates compared to gene-based and tumor microenvironment based biomarkers (so called GeneBio and TME-Bio, respectively). The gene based biomarkers defined as the expression levels of ICI targets including PD1, PD-L1, or cytotoxic T lymphocyte antigen 4 (CTLA4) and tumor microenvironment based biomarkers are defined as the levels of CD8 T cell, T cell exhaustion, cancer-associated fibroblast (CAF) and tumor-associated macrophage (TAM). Overall, the study is well designed and the conclusions are supported with the data. I list below the following points that needs to be addressed for further clarification of the soundness of the study.

1. The results can use a brief overview of the pathways they have identified using the top scoring 200 genes after applying PageRank using the drug target as the seed. How many pathways (Netbio) satisfying $P_{adj} < 0.01$ were there? What were these pathways? It is also unclear which multiple hypothesis testing method was used of P-value adjustment. Were all the pathways used as features in the logistic regression classifier?
2. I believe a cutoff value of 0.5 from logistic regression output has been used for the plots on Fig 2. In addition to AUROC, it would be interesting to see AUPRC (are under precision-recall curve) as well as the distribution of output probability values from the classifier between the observed responders and non-responders.
3. The authors express the importance of robustness of the biomarkers and argue that Netbio based biomarkers are more robust than gene-based biomarkers based on the random subselection of the training data set. However, the pathway-based biomarkers (Netbio) seem to depend significantly on the protein interaction network used (STRING score > 700). Given that our knowledge on known protein interactions is incomplete, to see the effect of such incompleteness, I would be curious to see whether these pathways are conserved if a smaller network, say STRING score > 900 . Would the identified Netbio still achieve improvement across GeneBio and TMEBio?
4. For the across-study performance evaluation (Fig 3), the three datasets were used separately to check the AUC, what would be the AUC (and AUPRC) on the datasets combining these 3 datasets? Can the authors devise a way to merge these datasets accounting for batch effects?
5. Like the point #4 above, can the 3 TCGA data sets be merged for correlation analysis (Fig 5)? The melanoma cohorts (Gide and Liu) used in training can also be merged to identify possible more universal pathway-based biomarkers. This is especially relevant as there does not seem to be a substantial overlap among the top 10 most important features identified from the two datasets. The small effect size in the correlation (significant but rather weak, i.e., $|PCC| < 0.4$) also deserves a discussion. Does it imply existence of certain subtypes of the disease among the individuals of the cohort for instance (perhaps the meta-data from these cohorts can hint some insights)?
6. To show improvement over using only TME-Bio, the authors combine those biomarkers with Netbio and present the survival rates for the combination and TME-Bio. It would be good to see what the survival rates for Netbio only are to check whether including TME-Bio makes sense or not.
7. Some methodological clarification would help for the following: (i) "gene/pathway expression levels are z-score-standardized", I have not been able to find details on how this was done, were the values for each gene/pathway standardized across the samples of the same cohort? (ii) what is the final value of w

that was identified after optimization across the parameter space? (iii) what was the dumping factor in PageRank algorithm used in network propagation (currently the text states that default parameters of the package were used but please specify as the implementation might change / be updated in the future).

8. Overall, it is not clear what would be the overall Specificity, Sensitivity and Precision for a classifier that uses Netbio for ICIs across 3 different cancer types? Perhaps this can be included in the discussion. Furthermore, a supplementary table containing the list of genes in each of the Genebio, TME-Bio and Netbio would be certainly useful for further validation / exploitation of the study findings.

Minor:

Pg3:49 must be developed => is needed

Pg3:57 many studies have reported => authors cite only 1 study

Pg15:313 $P < 10^{-16}$ => $P < 10^{-15}$ (given that the lowest P was 1.5×10^{-16})

Pg16:353-Pg17 raf activation => Raf activation (multiple occurrences)

Reviewer #4 (Remarks to the Author): Expert in immunotherapy, pathology, and artificial intelligence

In their manuscript „Network-based machine learning approach to predict immunotherapy response in cancer patients“, Kong and colleagues describe a machine learning framework which they call NetBio and use this to identify novel predictive biomarkers for the response to immune checkpoint inhibitors. To use network biology is an interesting approach to address this important clinical challenge and the experiments are rather comprehensive. However, I have various major concerns:

Methodology

- In the first experiments (Figure 2) the authors compare their NetBio model with the expression levels of drug targets such as PD-1, PD-L1, and CTLA4. While I know that this is a heavily debated issue, in my opinion this is not sufficient as it stops at the transcriptional / translational level. Post-translational modifications, protein-protein interactions, intra- and extracellular localization, and most importantly tumor heterogeneity are all not considered. However, scoring systems, which are currently used in a clinical setting (such as TPS, CPS, and IC score) are based on immunohistochemistry (IHC) to determine

exactly, which cells and tissues express these targets. It would be highly desirable to compare the NetBio model to these scores in the cohorts used.

- Furthermore, the authors claim, that they compared the NetBio model to two other network-based models called GeneBio and TME-Bio. However, throughout the manuscript, the comparisons are done for each “target” (PD1, PD-L1, CTLA4, CD8, T exhaust, etc.) individually. The authors should try training additional logistic regression models, which include all GeneBio and TME-Bio targets respectively. This would be a much “fairer” comparison.

- Additionally, there are more “sophisticated” models than l2-regularized logistic regression. The authors should consider trying out other methods such as SVMs, random forests, or even neural network based approaches.

Performance

- While there seems to be an improvement of the predictive performance of the NetBio model when compared to other transcriptomic features (PD1, PD-L1, etc.), the achieved values are generally rather low. This is even more worrying, as the number of cases is somewhat small (see l. e. ROC curve in Figure 3 b) and the class distribution is not directly obvious. The AUROC can overestimate the performance of a model, especially if the number of responders / non-responders is not balanced. Precision-Recall-Curves (with the indication of the class weight) and cross tables could help to better understand the performance of each approach.

Consequently, the ranks displayed in Figure 4 are not really meaningful to me, as a) the comparison is biased towards the NetBio approach (see comment above) and b) the performance even of the best ranking model is not really high.

Novelty

- In Figure 5 the authors show that there is a strong overlap between known predictors of ICI response and the NetBio model. This is unsurprising to me as a) TMB is such a basic and general parameter and b) ICI responses are so heavily dependent on various immune cell populations. The connection to FGFR signaling is interesting but not further explored.

Clinical applicability

- To me the most important aspect is how the NetBio model compares to or could be included in existing ICI prediction algorithms used in a clinical setting. As discussed above, most of these are based on IHC to account for tumor heterogeneity and morphology. In Figure 6 the authors claim consistency with IHC-

based immune phenotype in bladder-cancer. However, it remains completely unclear how this was determined (no mention of IHC in the methods section of the paper at all). Examples of the IHC stainings used would be highly desirable. Furthermore, the box plots in Figure 6 b and c are curious. Where does the p-value of 8.21×10^{-17} (!) come from? The error bars show considerable overlapping for a value this small. During ANOVA, did the authors account for a) the values distribution? b) the values standard deviation? c) multiple comparisons?

- The data on the addition of the NetBio model together with TMB in Figure 7 is interesting. However, while there was an initial hype in using TMB as a biomarker for ICI response, the importance of this parameter in a clinical setting has dramatically decreased recently. So, the performance increase would have to be really pronounced to include the use of both outside of a research setting.

- The determination of the ground truth, meaning the identification of responders and non-responders is a key aspect of this study and can pose a serious challenge. The authors write, that RECIST criteria were used in some of the patients. How many? Then the authors state: "For datasets that did not provide or use RECIST criteria, we used responder and non-responder classification from the datasets.". What exactly is meant by this statement? For how many patients was this "method" used to determine the ground truth (response vs. non-response)?

Other aspects

- There is no section that summarizes the statistical methods used in this manuscript. There is no statement on whether the underlying code can be accessed (i.e., within a GitHub repo, etc.). Furthermore, there is no statement on any ethics approval or that no ethics approval was necessary.

Reviewer #1 (Remarks to the Author): Expert in network medicine

--In that ~30% of patients respond to immune checkpoint inhibitor therapy, I suggest you change the phrase “only a few patients respond to immunotherapy...” to “the minority of patients respond to immunotherapy....”

Following the reviewer’s suggestion, we have changed the manuscript at page 2, line3 and page 3, line 7.

--How different were the gene expression patterns for network-based binding partners to the ICI drug targets among the different cancers? Would there be any (at least statistical) value in combining all cancers in one comprehensive data set and repeating the analysis?

We thank the reviewer for this comment. We used the gene expression levels of the network neighbors of ICI targets (PD1, PD-L1, and CTLA4) to measure transcriptome similarity between cohorts. We defined the transcriptome similarity as follows: (1) for each patient, we computed Spearman rank correlation to all patients in another cohort; (2) we took the maximum value from the correlation measurements; (3) we iterated steps (1) and (2) for all patients in both cohorts. We found that transcriptome similarity was high between the two melanoma cohorts (median transcriptome similarity of 0.39 and 0.41 for ICI responders and non-responders, respectively), whereas it was low between cohorts with different cancer types (Supplementary Fig.27), with median transcriptome similarities ranging from 0.14 to 0.26. This suggests that there is cancer type-specificity for the gene expression patterns.

Combining all cancers in one comprehensive dataset did not improve the performance of NetBio-based predictions (Supplementary Fig.28). We used ComBat to remove batch effects between cohorts and found that when four different cancer datasets (Gide, Liu, Kim, IMvigor210) were combined into a single dataset, the performance of LOOCV using NetBio based on the combined dataset was decreased compared with that using NetBio based on the individual datasets (Supplementary Fig.28). This suggests that expression biomarkers of ICI treatment response may differ across cancer types.

The above results have been updated in our manuscript on page 21, line 20.

Supplementary Figure 27. Gene expression similarity between different cohorts. (a)-(b) Transcriptome similarity among the Gide (melanoma), Liu (melanoma), Kim (metastatic gastric cancer), and IMvigor210 (bladder cancer) cohorts. Spearman correlation was used to measure transcriptome similarity. Median transcriptome similarity from pairwise correlation is shown, where (a) ICI responders and (b) ICI non-responders were used to compute the similarity. (c)-(n) Distribution of computed transcriptome similarity between two cohorts. Transcriptome similarity was measured using (c)-(h) ICI responders and (i)-(n) ICI non-responders. Black and red dotted lines denote transcriptome similarity at 0 and at the median of the distribution, respectively.

Supplementary Figure 28. Leave-One-Out Cross-Validation (LOOCV) predictive performance using a combined dataset of all cancer types. (a) Overall scheme for removing batch effects and LOOCV analysis. (b)-(c) Comparison of the overall predictive performance of NetBio markers and NetBio combined markers using (b) accuracy or (c) F1 score. (d)-(k) Predictive performance in each cohort using (d)-(g) accuracy or (h)-(k) F1 score.

--Have you considered developing a strategy for weighting the interaction network edges based at least on the connectivity of the drug target and of each of its binding partners? In the absence of data on protein concentrations and association constants for each binding partner, greater connectivity might be expected to correlate inversely with efficacy if, for example, only one or a few binding partners drive the primary response phenotype, as the other binding partners ‘dilute’ the effect by competition.

We thank the reviewer for this excellent comment. To measure ICI efficacy for each binding partner, we used patients’ gene expression level of the binding partner and ICI

response and computed an AUC to define ICI efficacy. We observed that in four different ICI-treated cohorts, ICI efficacy did not correlate with the connectivity of the binding partners (Supplementary Fig.31; P value < 0.05 considered significant). These results suggest that a gene's degree centrality is not a confounding factor when predicting ICI response.

The above results have been updated in our manuscript on page 27, line 10.

Supplementary Figure 31. Correlation between network degree of ICI drug targets (PD1, PD-L1, and CTLA4) and ICI efficacy. (a)-(d) Correlation between drug efficacy, determined by area under the receiver operating characteristic curve (AUC) based on gene expression levels and ICI response, and degree of network centrality. Correlation was computed using Spearman's rank correlation. The 95% confidence interval is shown for the linear regression line.

--I suggest that Figure S3 (or at least S3b) be included in a figure in the main text.

Following the reviewer's suggestion, we have moved the figure to the main text (Figure 4).

--Do you have any transcription profiles from tumors after treatment? If so, could you

validate the putative pathways predicted to govern response by analyzing changes in gene expression with treatment?

We thank the reviewer for this comment. Using a melanoma cohort (Riaz *et al.*), we identified differentially expressed pathways (DEPs) by comparing pathway expression prior to ICI treatment and during ICI treatment (Supplementary Fig.32a). We found that compared to all pathways available from the Reactome database, the NetBio pathways were not enriched with DEPs (Supplementary Fig.32b; two-sided Fisher's exact test $P = 0.5$). This result suggests that the expression levels of NetBio pathways do not necessarily change during treatment.

To further investigate the association between DEPs and drug response, we tested whether DEPs can accurately predict responders and non-responders in various melanoma datasets (Supplementary Fig.32a). We found that DEPs were less predictive of ICI treatment response compared with NetBio markers (Supplementary Fig.32c-p). Specifically, we used the expression profiles of DEPs to train a machine-learning model and conducted (i) within-study predictions (LOOCV) and (ii) across-study predictions (Supplementary Fig.32c,h). We observed that in some cases, DEPs provide information to differentiate responders from non-responders; however, in most cases, NetBio-based predictions were better than DEP-based predictions (Supplementary Fig.32c-p). These results suggest that the baseline gene expression profiles associated with drug response may not necessarily change after ICI treatment.

The above results are now included in the manuscript (page 27, line 18 and page 29, line 20).

Supplementary Figure 32. Identification of differentially expressed pathways (DEPs) prior to and during ICI treatment, and predictive performance based on expression levels of DEPs. (a) Overall scheme for identifying DEPs and conducting DEP-based ICI response prediction. Mann-Whitney U test was used to identify DEPs ($P < 0.05$ considered significant). (b) Enrichment of DEPs in NetBio pathways. Two-sided Fisher's exact test was used to measure statistical significance. (c)-(g) Predictive performance of within-study cross validation in the Gide and Liu datasets using (d)-(e) accuracy and (f)-(g) F1 score to quantify predictive performance. (h)-(p) Across-study predictive performance. The Gide dataset was used to train a machine-learning model, and the Auslander, Huang, Prat, or Riaz datasets were used to test the predictive performance. To quantify performance, we used (i)-(l) area under the receiver operating characteristic curve (AUC) and (m)-(p) area under the precision-

recall curve (AUPRC).

--Please comment on the potential benefits of using single-cell RNAseq analysis as transcriptomic data sources (as perhaps a next step). Using mean-field values for gene expression, especially in cancers, runs the risk of failing to detect a small population of cells (or of gene products) that primarily drive the response. The individual cell-based transcriptome or a limited population of cells whose transcriptomes are sufficiently similar may offer the opportunity for even better predictive accuracy if the analysis were also coupled with network-based information.

We thank the reviewer for this interesting suggestion. Single-cell RNA seq data would likely improve the performance of network-based machine-learning approaches, but that would require further sequencing of primary tumors, which is out of scope of this study. Following the reviewer's suggestion, we have included the following comment in our Discussion section (page 21, line 15):

“Given the consistency of our results, a future research opportunity would be to apply the network-based approach with higher-resolution sequencing techniques (e.g. single-cell RNA sequencing data) that enable consideration of important aspects of the immune microenvironment, including immune cell proportions or cell states.”

Reviewer #2 (Remarks to the Author): Expert in immunotherapy response and biomarkers in melanoma

The authors have submitted a manuscript analyzing a curated dataset of 700 ICI treated patients with available clinical outcomes and transcriptomic data

Used NetBio predictions in melanoma, gastric cancer, bladder cancer
Compared to “other ICI treatment biomarkers, such as ICI targets”

“Presents a network-based method to effectively select IO-response-associated biomarkers that make robust ML-based predictions for precision oncology”

In the introduction, the statement: “programmed cell death 1 (PD1)/programmed cell death-ligand 1 (PD-L1) expression by immunohistochemistry is a Food and Drug Administration (FDA)-approved companion diagnostic test for various cancer types. REF: 4”

- Unfortunately, melanoma (one of the cancer types selected in this manuscript) is not one of those types. So the utilization of PD-L1 as a prognostic biomarker in melanoma does not make sense and undermines the subsequent analysis.

- References mentioned 4, 6-8 in PD-L1 underperformance are also not in melanoma.

Would agree that better biomarkers needed. Question remains why the comparison of performance is then compared to these clearly underperforming biomarkers (which again in the case of melanoma is not used clinically at all).

We agree with the reviewer that our NetBio-based machine-learning model can be tested against better-performing biomarkers. We first compared with ICI-target-based markers, because NetBio pathways are discovered from the network-neighbors of ICI targets. To test NetBio against better-performing biomarkers, we compared it with other state-of-the-art immunotherapy response prediction methods (PMID: 30127393, PMID: 34430923, PMID: 34019806, PMID: 30127394), as well as with a deep neural network-based method (PMID: 31825821). We first tested the predictive performance for leave-one-out cross-validation. We found that NetBio-based predictions were better in 33 of 34 comparisons (Supplementary Fig.10; 97.1%). Also, in across-study predictions, we observed that NetBio-based predictions were better in 17 of 18 comparisons (Supplementary Fig.11; 94.4%). These results suggest that NetBio can improve prediction of ICI treatment responses compared with other biomarkers.

We have included the additional results in the manuscript on page 12, line 4.

Supplementary Figure 10. Comparison of Leave-One-Out Cross-Validation (LOOCV) performances. (a) Overall scheme of immunotherapy response prediction using LOOCV. (b)–(c) Summarized classification results using (b) accuracy and (c) F1 score as a metric to quantify predictive performance. (d)–(k) LOOCV predictive performances based on (d)–(g) accuracy and (h)–(k) F1 score.

Supplementary Figure 11. Comparison of across-study predictive performance. (a) Overall scheme of across-study predictions of immunotherapy response. (b)–(c) Summarized prediction results using (b) area under the receiver operating characteristic curve (AUC) and (c) area under the precision-recall curve (AUPRC) as a metric to quantify predictive performance. (d)–(k) Across-study predictive performance based on (d)–(g) AUC and (h)–(k) AUPRC.

Would also agree that network based models have the potential to outperform others, but in doing this comparison combining multiple disparate disease types may be a challenge in data interpretation.

We agree with the reviewer that combining different cancer types can be a challenge. In our original manuscript, we did not combine different cancer types in order to avoid challenges in data interpretation. We have updated our manuscript for a clearer understanding

of our analysis (page 24, line 19).

NetBio created by:

- Used STRING PPI network using ICI targets as seed genes
- Selected genes with high influence scores (top 200)
- Used Reactome to identify pathways enriched with the genes
- Used the pathways (not genes) to predict ICI responses

After creating NetBio

- Negative control: gene based biomarkers (IO target genes), TME genes, or pathways from ML approaches
- Compared to: 1: drug response measured by reduced tumor size after ICI tx or 2: OS
o Is (1) RECIST or not?

RECIST criteria-based responses were used for the Liu, Gide, IMvigor210, Kim, Riaz, and Prat data. We added a supplementary table to clarify the drug-response labels that we used throughout the manuscript (Supplementary Table 1).

Cohort name	Cancer type	Drug	Drug response label	Within or across study analysis
Liu	melanoma	Nivolumab, Pembrolizumab	RECIST	within study, across study
Gide	melanoma	Pembrolizumab, Nivolumab, Ipilimumab	RECIST	within study, across study
IMvigor210	bladder cancer	Atezolizumab	RECIST	within study
Kim	gastric cancer	Pembrolizumab	RECIST	within study
Auslander	melanoma	anti-PD-1, anti-CTLA-4	responder / non-responder	across study
Riaz	melanoma	Nivolumab	RECIST	across study
Prat	melanoma	Nivolumab, Pembrolizumab	RECIST	across study
Huang	melanoma	Pembrolizumab	recurrence	across study

Supplementary Table 1. Cohorts and drug response labels used in this study. For cohorts that used RECIST criteria, we considered patients with Complete Response (CR) or Partial Response (PR) as responders and those with Stable Disease (SD) or Progressive Disease (PD) as non-responders.

o For RECIST data, methods state aggregation: CR/PR and SD/PD – SD patients are often actually considered responders since they are non-progressing

We selected CR and PR as responders and SD and PD as non-responders, as was done in previous studies including Lee *et al.* (PMID: 33857424), Ding *et al.* (PMID: 27354694), Geeleher *et al.* (PMID: 24580837), Majumder *et al.* (PMID: 25721094), Sakellaropoulos *et al.* (PMID: 31825821), and Sharifi-Noghabi *et al.* (PMID: 32657371). Considering the reviewer's suggestion, we conducted an experiment to predict progression-free survival (PFS). We found that the NetBio-based machine-learning model could

accurately predict PFS (Supplementary Fig.1), suggesting that our machine-learning model can distinguish progressing tumors from non-progressing tumors.

Supplementary Figure 1. Prediction of progression free survival (PFS) in (a) Gide and (b) Liu dataset.

o Other datasets just used their R vs NR status (did that agree with CR/PR and SD/PD categories)? Should be the same across all data sets (and often these variable do differ across studies)

For the Liu, Gide, IMvigor210, Kim, Riaz, and Prat datasets, RECIST criteria-based labels were used to classify responders and non-responders. For RECIST datasets, we uniformly considered patients with CR and PR as responders, whereas we considered patients with SD and PD as non-responders. In previous publications, such as PMID: 30127394 and PMID: 34430923, the R vs. NR status of the Auslander dataset was analyzed along with RECIST criteria-based datasets. It would be interesting to test the extent to which drug response labels in different studies are comparable with each other; however, that research is out of the scope of this study.

o Also OS is not a great marker for response to IO (patients will see many other lines of treatment), usually use PFS or even cancer-specific survival

We thank the reviewer for this comment. We found that NetBio-based prediction of progression free survival (PFS) was accurate in two independent datasets. Specifically, when using NetBio-based LOOCV prediction, patients predicted to be ICI responders in the Gide and Liu datasets had prolonged PFS compared with patients predicted to be non-responders (Supplementary Fig.1 a,b; log-rank test $P < 0.05$ considered significant). By comparison, we found that drug target-based prediction of PFS was less consistent (Supplementary Fig.1 a,b). In particular, prediction based on PD1 expression in the Liu dataset was inversely predictive of PFS (Supplementary Fig.1b). These results suggest that NetBio-based prediction of ICI

response is correlated with PFS as well as overall survival.

We have added the new results to the manuscript (page 8, line 4).

- Training and validation from same data set
- Used 2 melanoma cohorts and 1 bladder cohort to measure performance

Feedback:

Overall, the authors bring together 700 quite heterogeneous datasets including different tumor types (unclear why these types chosen) with different immune checkpoint inhibitor treatments. The analysis moves back and forth from different datasets and it is unclear how data sets were integrated or chosen for analysis (see specific comments on melanoma – Liu et al. included in some places and not in others). These inconsistencies are quite troubling.

We agree with the reviewer and have updated our manuscript to improve the consistency of our results. We used the Liu data to train a machine-learning model and tested the predictive performance in three independent cohorts (Supplementary Fig.7a). We used AUC as a metric to measure predictive performance and observed that NetBio-based predictions outperformed other ICI-related biomarkers in 88.5% (23/26) of all comparisons (Supplementary Fig.7b-d). In our original manuscript, we used the Gide dataset to train the machine-learning model, because the Gide dataset resulted in better LOOCV prediction performance compared with the Liu dataset. These results suggest that regardless of the training datasets used to train the machine-learning model, NetBio-based markers can improve prediction performances compared with drug target-based or tumor microenvironment-based markers.

We have updated the results in the manuscript accordingly (page 10, line 22).

Supplementary Figure 7. Predictive performance in three independent melanoma datasets. (a) Overall scheme of immunotherapy response prediction in three independent datasets. The Liu dataset was used to train a machine-learning models. (b)-(d) The area under the receiver operating characteristic curve (AUC) for the (b) Auslander, (c) Prat, and (d) Riaz datasets is shown. The random expectation, equaling an AUC of 0.5, is displayed as dotted lines. No expression profiles of cancer-associated fibroblast (CAF) marker genes were available in the Prat dataset. N.D., Not detected.

Additionally, the efforts to bridge across different tumor types is quite challenging and makes data interpretation difficult. Beyond melanoma, there are certainly other published data sets in each of the tumor types listed for validation. Why not included here?

We agree with the reviewer that interpretation of results generated by combining different tumor types is difficult. We note, however, that we did not integrate different cancer types when making predictions. We have updated our manuscript to better clarify that our predictions were done for individual cancer types and not for combined cancer types (page 24, line 19).

While we hope to test our prediction model in additional independent cancer cohorts, publicly accessible data for ICI-treated patients with bladder cancer or gastric cancer are

currently not available. On the other hand, we tested the predictive performance of the NetBio-based machine-learning model in an additional cohort of anti-PD1-treated melanoma patients (Supplementary Fig.8a) and found that the NetBio-based machine-learning model could accurately predict ICI-responses (Supplementary Fig.8b, c). Specifically, we used the Gide or Liu datasets to train NetBio-based machine-learning models and then tested the models' predictive performance using the Huang (PMID: 30804515) dataset. As the Huang dataset included recurrence of cancer in anti-PD1-treated melanoma patients, we labeled patients with no cancer recurrence as ICI responders and patients with cancer recurrence as ICI non-responders. When using AUC as a measure of predictive performance, we found that regardless of the training dataset used, the NetBio-based machine-learning model was able to accurately predict ICI responders and non-responders (Supplementary Fig.8b, c; Gide to Huang AUC = 0.78, Liu to Huang AUC = 0.8). These results suggest that NetBio-based machine-learning models can be a useful framework when predicting ICI responses in new datasets. The predictions for the new melanoma cohort are now included on page 11, line 7 of the manuscript.

Supplementary Figure 8. Performance of across-study predictions in an additional cohort. (a) Overall scheme of immunotherapy response prediction in an independent dataset. (b)-(c) The area under the receiver operating characteristic curve (AUC) for (b) Gide to Huang and (c) Liu to Huang. The random expectation is displayed as dotted lines.

In this vein, subsequent integration with TCGA to “validate IO responses” is unfounded. SKCM TCGA (and likely others) did not include patients on systemic therapies as these were “landscape biology” efforts. The authors attempt to correlate their signatures/predictions with features in the TCGA, but none of which make any sense. In SKCM TCGA, they correlate with Ag presentation (in one dataset) and FGF (in another). There is no effort to align with the actual TCGA phenotypes, one of which is “immune” suggesting their data may not align with known biology.

We thank the reviewer for this comment. We predicted ICI responses in TCGA cancer patients to understand whether our predictions are in line with immune contextures that are already known to be associated with ICI response. We chose the TCGA dataset because immune contextures are well studied in the TCGA dataset. Following the reviewer’s

suggestion, we also compared the NetBio-based predictions to the TCGA phenotypes. We found that regardless of the training dataset used (Gide or Liu), patients predicted to be responders by the NetBio-based models were highly likely to have the “immune” phenotype in the SKCM TCGA data (Supplementary Fig.14). Although the NetBio-based predictions were highly correlated with immune infiltration regardless of which data were used to train the machine-learning model, the biological pathways that were selected as important were different (Ag presentation and FGFR signaling when using Gide and Liu data, respectively). This result suggests that there are different modes of immune infiltration in melanoma. In line with our results, a recent study reported that mouse models that lack FGF2, a member of the FGFR signaling pathway, were more inflammatory than mouse models with intact FGF2 (PMID: 32792542). These results suggest that NetBio can capture infiltration of immune cells by using gene expression levels.

The result is now updated in the manuscript (page 14, line 11).

Supplementary Figure 14. Association between predicted ICI response and TCGA subtypes in melanoma patients. (a)-(b) Associations measured using the (a) Gide or (b) Liu cohort to train a NetBio-based machine-learning model. Statistical significance was measured using Mann-Whitney U test. Boxplot shows median value, interquartile range (IQR) as bounds of the box and whiskers that extends from the box to upper/lower quartile \pm IQR \times 1.5.

After creating NetBio

-Negative control: gene based biomarkers (IO target genes), TME genes, or pathways from ML approaches

-Compared to: 1: drug response measured by reduced tumor size after ICI tx or 2: OS

Is (1) RECIST or not?

For RECIST data, methods state aggregation: CR/PR and SD/PD – SD patients are often actually considered responders since they are non-progressing

As was done in previous studies (PMID: 33857424, PMID: 27354694, PMID:

24580837, PMID: 25721094, PMID: 31825821 and PMID: 32657371), we considered patients with CR and PR as responders and patients with SD and PD as non-responders. To consider tumor progression in predicting response to ICI treatment, we now show that NetBio-based machine-learning models can accurately predict progression-free survival (Supplementary Fig.1).

Supplementary Figure 1. Prediction of progression free survival (PFS) in (a) Gide and (b) Liu dataset.

Other datasets just used their R vs NR status (did that agree with CR/PR and SD/PD categories)? Should be the same across all data sets (and often these variable do differ across studies)

For datasets with RECIST criteria-based drug response labels, including the Liu, Gide, IMvigor210, Kim, Riaz, and Prat datasets, we uniformly considered patients with CR and PR as responders and patients with SD and PD as non-responders.

Also OS is not a great marker for response to IO (patients will see many other lines of treatment), usually use PFS or even cancer-specific survival

We found that NetBio-based machine-learning models could accurately predict PFS in the Gide and Liu datasets (Supplementary Fig.1), whereas drug target-based predictions were less consistent in their predictions (Supplementary Fig.1).

In Fig. 6 the authors switch to another TCGA and use IHC to compare “excluded” and “immune included” tumors – an abrupt transition with no real explanation.

We thank the reviewer for this comment. We have updated our manuscript to include the explanation of our results (page 16, line 7).

Finally, Fig. 7 looks only at bladder cancer and integrates TMB.

We thank the reviewer for this comment. In addition to the bladder cancer cohort (IMvigor210), we found that combining NetBio and TMB improved ICI response prediction in a melanoma cohort. Specifically, we used datasets that provide both TMB levels and overall survival data (the Liu cohort for melanoma and the IMvigor210 cohort for bladder cancer). We used z-score-standardized TMB levels and NetBio pathway expression levels for machine-learning training. For the Liu dataset, the number of nonsynonymous mutations was used as the TMB level, according to the original paper (PMID: 31792460). Log-rank tests showed that prediction of overall survival in the Liu dataset was more accurate when NetBio and TMB levels were used (Fig.S29c; $P = 4.61 \times 10^{-3}$) compared with when just TMB levels were used (Fig.S29a; $P = 2.7 \times 10^{-2}$), suggesting that NetBio-based transcriptomic features contain ICI-related signatures that are distinct from TMB levels.

The above results are now included on page 22, line 18.

Supplementary Figure 29. Prediction of overall survival in the Liu dataset. Prediction using (a) TMB, (b) NetBio, or (c) a combined model (TMB + NetBio). Statistical significance was measured using log-rank test.

Summary:

Overall, despite interesting methodologies and approaches, there is little consistency and synergy of data analysis and validation across the different data sets and tumor types. This undermines the real-world applicability of this type of analysis.

I would recommend the authors narrow to one histology, enlarge the included datasets and align more closely with the known phenotypes in that field. The current all-inclusive

approach increases the total “n” of samples, but likely blurs what is likely distinct biology.

As suggested by the reviewer, to improve the consistency of our analysis, we have included (i) across-study predictions using the Gide (Figure 3) and Liu datasets (Supplementary Fig.7) and (ii) NetBio-based and TMB-based predictions in an additional melanoma cohort (Supplementary Fig.29). Moreover, to test the consistency of the predictive performance, we measured the predictive performance of NetBio-based machine-learning using area under the precision-recall curve (AUPRC). These results show that NetBio consistently outperforms other ICI-target-based or TME-based markers (Supplementary Fig.5). Also, additional comparisons with state-of-the-art ICI-response prediction methods and a deep learning-based approach consistently show that NetBio can improve predictive performance (Supplementary Fig.10, Supplementary Fig.11). All of the results suggest that NetBio-based machine-learning can consistently predict ICI response.

To align our prediction results with known phenotypes, we measured correlations between NetBio-based machine-learning predictions and TCGA phenotypes in a melanoma cohort (Supplementary Fig.14). We found that regardless of the dataset (Gide or Liu) used to train the machine-learning model, the predicted probability of positive response to ICI was highest for patients with the “immune” TCGA phenotype (Supplementary Fig.14), suggesting that our predictions align with immune infiltration levels in melanoma patients.

Specific comments from manuscript:

Across-study predictions:

-Here used Gide et al to train and Auslander, Prat, and Riaz to test performance

-Why was Liu et al excluded here? It was used previously as a representative melanoma dataset

Following the reviewer’s suggestion, we found that when using the Liu dataset to train the machine-learning model, NetBio outperforms markers based on drug targets or the tumor microenvironment (Supplementary Fig.7). Specifically, we found that NetBio-based predictions outperformed other ICI-related biomarkers in 88.5% (23/26) of all comparisons (Fig S7b-d). This result suggests that NetBio-based predictions can improve predictive performance regardless of the training dataset used.

Supplementary Figure 7. Predictive performance in three independent melanoma datasets. (a) Overall scheme of immunotherapy response prediction in three independent datasets. The Liu dataset was used to train a machine-learning models. (b)-(d) The area under the receiver operating characteristic curve (AUC) for the (b) Auslander, (c) Prat, and (d) Riaz datasets is shown. The random expectation, equaling an AUC of 0.5, is displayed as dotted lines. No expression profiles of cancer-associated fibroblast (CAF) marker genes were available in the Prat dataset. N.D., Not detected.

NetBio based predictions recapitulate immune microenvironment in external TCGA datasets
 - Here tested how NetBio-based predictions correlated with immune contextures in TCGA
 - Used Gide or Liu dataset to predict ICI response in melanoma patients in TCGA dataset (therapeutic response is NOT present in TCGA dataset)

Immune contextures are well studied in the TCGA dataset. Therefore, we tested if our predictions correlated with immune cell proportions known to be associated with ICI response.

Went on to look at gastric and bladder cancer.

Finally, the group combined NetBio with TMB in aPD1 treated bladder cancer.

We investigated combining NetBio with TMB because TMB is widely accepted as an important biomarker to predict ICI response, and responders and non-responders often share similar TMB levels, suggesting the need for additional biomarkers. We found that in bladder cancer (Figure 7) and melanoma (Supplementary Fig.29), combining NetBio and TMB improved ICI response predictions. These results suggest that NetBio captures ICI response-related signatures that are distinct from TMB.

Major issues:

Not all immune checkpoint inhibitors are the same, biomarkers likely different. This is reflected in analysis in the Liu et al paper (patients previously tx with aCTLA4 vs aPD1 naive patients) and different histologies:

Melanoma

- Gide et al. (nivo-, pembro-, and/or Ipi)
- Liu et al. (nivo- or pembro- (some previously treated with aCTLA4, subanalysis in this paper))
- Auslander et al (aPD1 and/or anti-CTLA4)
- Prat et al (nivo-, pembro-)
- Riaz et al (nivo-)

Gastric

- Kim et al (pembro-)

Bladder

- IMvigor210 (atezo-)

Identifying biomarkers for each ICI is important, and more ICI cohorts are required to help identify novel biomarkers. Because NetBio considers network neighbors of ICI targets, it is possible that NetBio-based machine-learning captures tumor microenvironment-associated signatures, enabling consistent predictive performance across different ICI treatments. Indeed, NetBio-based predictions were correlated with immune cell proportions (Figure 5). Moreover, TCGA melanoma patients with the immune phenotype were more likely to be predicted as ICI responders than patients with other phenotypes when NetBio was used to train a machine-learning model (Supplementary Fig.14). These results suggest that NetBio-based predictions recapitulate aspects of the tumor microenvironment that are associated with ICI response.

Supplementary Figure 14. Association between predicted ICI response and TCGA subtypes in melanoma patients. (a)-(b) Associations measured using the (a) Gide or (b) Liu cohort to train a NetBio-based machine-learning model. Statistical significance was measured using Mann-Whitney U test. Boxplot shows median value, interquartile range (IQR) as bounds of the box and whiskers that extends from the box to upper/lower quartile \pm IQR \times 1.5.

Major Comments on Figures:

-Fig 2A/B, E/F: Why compare to PD1, CTLA4 in Gide and only PD1 in Liu? Makes no sense.

We used PD1 and CTLA4 in the Gide dataset because the Gide cohort contains patients who were treated with anti-PD1 and anti-CTLA4, whereas only anti-PD1 was used in the Liu cohort. Following the reviewer’s comment, we measured the performance of PD1 and CTLA4 gene expression profiles to predict drug responses in the Liu dataset. We conducted LOOCV in the Liu dataset and found that predictions based on PD1 and CTLA4 were inaccurate (Supplementary Fig.2a; Fisher’s exact test $P = 0.57$). Moreover, we found that predictions based on PD1 and CTLA4 were not able to accurately predict overall survival and progression-free survival (Supplementary Fig.2 b-c; log-rank test $P > 0.05$). These results suggest that ICI target-based gene expression profiles are not a great predictor of ICI response.

The above results are included on page 8, line 8.

Supplementary Figure 2. Prediction of drug response, overall survival (OS), and progression free survival (PFS) in the Liu dataset using expression levels of PD1 and CTLA4. (a) Immunotherapy response prediction using the expression levels of PD1 and CTLA4. The proportions of observed responders (teal) and non-responders (orange) among predicted responders (Pred R) and non-responders (Pred NR) are shown. Fisher's exact test was used to compute statistical significance. (b)-(c) The (b) OS and (c) PFS of predicted responders and non-responders based on LOOCV. The predicted responders and non-responders are depicted in red and blue, respectively. The log-rank test was used to measure statistical significance.

-Fig 3: why do the authors suddenly exclude the Liu data set and only train with Gide et al? Liu disappears from this analysis altogether
 -Fig 4a: Gide now again compared to Liu
 -Fig 4c: now Gide vs all (no Liu) – this back and forth makes no sense to me

We have updated the manuscript to improve the consistency. The results using the Liu data for training are now shown in Supplementary Figure 7.

Supplementary Figure 7. Predictive performance in three independent melanoma datasets. (a) Overall scheme of immunotherapy response prediction in three independent datasets. The Liu dataset was used to train a machine-learning models. (b)-(d) The area under the receiver operating characteristic curve (AUC) for the (b) Auslander, (c) Prat, and (d) Riaz datasets is shown. The random expectation, equaling an AUC of 0.5, is displayed as dotted lines. No expression profiles of cancer-associated fibroblast (CAF) marker genes were available in the Prat dataset. N.D., Not detected.

-TCGA has no response data (these were all untreated patients) – so the correlation of immune signatures with IO response is speculative at best. Also undermined by differences amongst the datasets, even in the same tumor type (e.g. melanoma) – Gide, Ag presentation; Liu, FGFR signaling.

-Also SKCM TCGA has clearly delineated “immune” subtype melanomas, no evidence of analysis or alignment with the published SKCM tumor groups.

Following the reviewer’s comment, we now show that NetBio-based machine-learning predictions are highly correlated with the “immune” subtype in melanoma patients, regardless of the dataset used to train the machine-learning model (Supplementary Fig.14).

-Fig 6: now compared to excluded and inflamed phenotypes using IHC. No consistency in criteria used across these different analyses.

IHC data from an ICI-treated cohort were only available for the bladder cancer (IMvigor210) dataset. It would be interesting to explore associations between immune infiltration levels and expression levels of NetBio pathways in various cancer types; however, additional sequencing of IHC and transcriptome data from ICI-treated cancer patients is required, which is out of the scope of this study.

-Fig. 7: suddenly focuses on just bladder cancer aPD-L1 treated patients and combination with TMB. Seems out of the blue for the paper.

We focused on combining NetBio-based predictions with TMB because although TMB is an important feature to predict ICI response, it is not a sufficient predictor of ICI response. Therefore, we tested whether combining NetBio-based transcriptomic features with TMB can improve ICI response prediction.

We have further clarified the rationale for our analysis on page 17, line 5.

Minor:

-Recommend an experimental overview/schema since including and aggregating data from a wide variety of studies (and integrated differently in different sections)

We agree with reviewer's comment about including an experimental overview. An experimental overview is now included in some of the main figures and supplementary figures (e.g. Figure 3-7). Additionally, we agree that an explanation of the wide variety of studies is required. We therefore added a supplementary table (Supplementary Table 1) that includes the cohorts used in this study, as well as the analysis conducted for each dataset.

Cohort name	Cancer type	Drug	Drug response label	Within or across study analysis
Liu	melanoma	Nivolumab, Pembrolizumab	RECIST	within study, across study
Gide	melanoma	Pembrolizumab, Nivolumab, Ipilimumab	RECIST	within study, across study
IMvigor210	bladder cancer	Atezolizumab	RECIST	within study
Kim	gastric cancer	Pembrolizumab	RECIST	within study
Auslander	melanoma	anti-PD-1, anti-CTLA-4	responder / non-responder	across study
Riaz	melanoma	Nivolumab	RECIST	across study
Prat	melanoma	Nivolumab, Pembrolizumab	RECIST	across study
Huang	melanoma	Pembrolizumab	recurrence	across study

Supplementary Table 1. Cohorts and drug response labels used in this study. For cohorts that used RECIST criteria, we considered patients with Complete Response (CR) or Partial Response (PR) as responders and those with Stable Disease (SD) or Progressive Disease (PD) as non-responders.

-Text in figures is fuzzy and hard to read

Following the reviewer's suggestion, we have updated the fuzzy text in Figure 5.

Reviewer #3 (Remarks to the Author): Expert in network medicine and systems biology

Kong and colleagues present a timely method to predict response to several immune checkpoint inhibitors (ICIs) used for melanoma, gastric cancer and bladder cancer. The method relies on building a logistic regression classifier using biological pathways that are enriched with genes that are in close vicinity of the drug targets in the human interactome. The classifier is trained and tested using data from the same cohorts (via leave-one-out cross-

validation) as well as using data from independent cohorts. When expression of genes in the biological pathways enriched with genes proximal to drug targets (so called Netbio – network based biomarkers) were used, the authors show modest but clinically significant improvement on the predicted responders (in around 90% of different comparisons conducted across data sets) and survival rates compared to gene-based and tumor microenvironment based biomarkers (so called GeneBio and TME-Bio, respectively). The gene based biomarkers defined as

the expression levels of ICI targets including PD1, PD-L1, or cytotoxic T lymphocyte antigen 4 (CTLA4) and tumor microenvironment based biomarkers are defined as the levels of CD8 T cell, T cell exhaustion, cancer-associated fibroblast (CAF) and tumor-associated macrophage (TAM). Overall, the study is well designed and the conclusions are supported with the data. I list below the following points that needs to be addressed for further clarification of the soundness of the study.

1. The results can use a brief overview of the pathways they have identified using the top scoring 200 genes after applying PageRank using the drug target as the seed. How many pathways (Netbio) satisfying $P_{adj} < 0.01$ were there? What were these pathways? It is also unclear which multiple hypothesis testing method was used of P-value adjustment. Were all the pathways used as features in the logistic regression classifier?

The number of NetBio pathways selected for the Gide, Liu, Kim, and IMvigor210 cohorts was 472, 323, 292, and 353, respectively. We now provide the NetBio pathways in the Source Data. For multiple hypothesis testing, we used the Holm-Sidak test. Also, we used the expression profile of all NetBio pathways to train a logistic regression classifier.

We have included the details in the Methods section (page 27, line 4).

2. I believe a cutoff value of 0.5 from logistic regression output has been used for the plots on Fig 2. In addition to AUROC, it would be interesting to see AUPRC (are under precision-recall curve) as well as the distribution of output probability values from the classifier between the observed responders and non-responders.

We thank the reviewer for this comment. We used the Gide dataset to train the machine-learning model and tested the predictive performance on three independent cohorts (Auslander, Prat, and Riaz). We found that both GeneBio-based predictions (average AUPRC = 0.47; Supplementary Fig.5) and TME-Bio-based predictions (average AUPRC = 0.33; Supplementary Fig.5) performed less optimal compared to NetBio-based predictions (average AUPRC = 0.63; Supplementary Fig.5). The probability distribution from the classifier is now included in Supplementary Fig.35. Also, we tested if NetBio-based prediction performed better than other methods when a smaller number of samples was used to train the machine-learning model. Specifically, for 100 different iterations using 80% of the randomly sampled patients from the training dataset (Gide dataset), NetBio-based prediction showed significantly better or equal performance in 26 of 27 comparisons using AUPRC (96.3%; Supplementary Fig.9). These results suggest that NetBio-based machine-learning can make robust predictions in comparison with predictions based on expression levels of other ICI-

response-associated biomarkers.

These results have been updated in the manuscript (page 10, line 16 and page 28, line 20).

Supplementary Figure 5. Predictive performance in three independent melanoma datasets. (a) Overall scheme of immunotherapy response prediction in three independent datasets. The datasets used to train and test the machine-learning models, transcriptomic features used to train the models, and number of samples for each dataset are displayed. (b)-(d) The area under the precision-recall curve (AUPRC) for the (b) Auslander, (c) Prat, and (d) Riaz datasets are shown. No expression profiles of cancer-associated fibroblast (CAF) marker genes were available in the Prat dataset. N.D., Not detected.

Supplementary Figure 35. Response probability of across-study predictions using a NetBio-based logistic regression model. (a)-(h) Response probability using the (a)-(d) Gide or (e)-(h) Liu data to train the machine-learning model. Boxplot shows median value, interquartile range (IQR) as bounds of the box and whiskers that extends from the box to upper/lower quartile $\pm IQR \times 1.5$.

Supplementary Figure 9. Prediction performances using fewer training samples to predict immunotherapy responses in external melanoma datasets. (a) Overall scheme for predictions in external melanoma datasets. (b)–(c) Summarized classification results (two-sided Student’s t-test; $P < 0.05$ was considered significant) using (b) AUC and (c) AUPRC as a metric to quantify prediction performance. (d)–(i) Boxplots of prediction performances in three different cohorts, using (d)–(f) AUC and (g)–(i) AUPRC as a metric to quantify prediction performance. Boxplot shows median value, interquartile range (IQR) as bounds of the box and whiskers that extends from the box to upper/lower quartile $\pm IQR \times 1.5$.

3. The authors express the importance of robustness of the biomarkers and argue that Netbio based biomarkers are more robust than gene-based biomarkers based on the random subselection of the training data set. However, the pathway-based biomarkers (Netbio) seem to depend significantly on the protein interaction network used (STRING score > 700). Given

that our knowledge on known protein interactions is incomplete, to see the effect of such incompleteness, I would be curious to see whether these pathways are conserved if a smaller network, say STRING score > 900. Would the identified Netbio still achieve improvement across GeneBio and TMEBio?

We found that the NetBio pathways were conserved when a smaller network (constructed with STRING score > 900) was used (Supplementary Fig.24). Specifically, we compared NetBio pathways found from STRING > 900 (NetBio 900) with to those found from STRING > 700 (NetBio 700). To measure the similarity between NetBio 900 and NetBio 700, we calculated an overlap coefficient (Supplementary Fig.24a). Across all four cohorts tested (Gide, Liu, Kim, and IMvigor210), we observed high overlap coefficients (Supplementary Fig.24b). These results show that the majority of the pathways in NetBio 900 are included in NetBio 700, suggesting that the pathway-based biomarkers are conserved in a smaller network.

Also, we found that while NetBio 900 showed reduced predictive performance compared with NetBio 700, the network-based approach was still effective in predicting ICI response (Supplementary Fig.25, Supplementary Fig.26). In a within-study leave-one-out cross-validation task, we found that NetBio 900 was equal to or better than other ICI biomarkers such as GeneBio and TME-Bio in 64 of 72 comparisons (Supplementary Fig.25; 88.9%). Moreover, in across-study predictions, NetBio 900 had better predictive performance in 40 of 54 comparisons (74.1%) when tested against other ICI biomarkers (Supplementary Fig.26). These results suggest that although using a smaller network affects the performance of ICI response prediction, the network-based approach still performs better than gene-based biomarkers. Also, the use of an incomplete network leads to reduced predictive performance, suggesting the importance of network coverage for identifying drug-response-associated biomarkers.

These results have been added to the manuscript (page 20, line 6).

Supplementary Figure 24. Testing for conservation of NetBio pathways between STRING > 700 and STRING > 900. (a) Overview of similarity calculation using overlap coefficients. (b) Similarity of STRING > 700 NetBio pathways and STRING > 900 NetBio pathways.

Supplementary Figure 25. LOOCV predictive performance using the STRING > 900 PPI network. (a) Overall scheme of immunotherapy response prediction using LOOCV. (b)-(c) Summarized classification results using (b) accuracy or (c) F1 score. (d)-(k) LOOCV predictive performance using (d)-(g) accuracy and (h)-(k) F1 score.

Supplementary Figure 26. Across-study prediction using the STRING > 900 PPI network. (a) Overall scheme of across-study immunotherapy response prediction. (b)-(c) Summarized classification results using (b) area under the receiver operating characteristic curve (AUC) or (c) area under the precision-recall curve (AUPRC). (d)-(j) Across-study predictive performances using (d)-(g) AUC and (h)-(j) AUPRC.

4. For the across-study performance evaluation (Fig 3), the three datasets were used separately to check the AUC, what would be the AUC (and AUPRC) on the datasets combining these 3 datasets? Can the authors devise a way to merge these datasets accounting for batch effects?

Following the reviewer's suggestion, we combined three different datasets to test the predictive performance. To combine different datasets while reducing batch effects, we conducted z-score standardization (PMID: 22851511) for each test dataset (Auslander, Prat

and Riaz) and then combined the standardized expression data to predict ICI responses (Supplementary Fig.6a). We observed that for both AUC and AUPRC, NetBio performed the best compared with other biomarkers (Supplementary Fig.6b-c), suggesting the robustness of our network-based approach.

These results are now included in the manuscript (page 10, line 18).

Supplementary Figure 6. Predictive performance in combined melanoma datasets. (a) Overall scheme of immunotherapy response prediction. (b)-(c) Predictive performance using (a) area under the curve (AUC) or (b) area under the precision-recall curve AUPRC. No expression profiles of cancer-associated fibroblast (CAF) marker genes were available in the Prat dataset. N.D., Not Determined.

5. Like the point #4 above, can the 3 TCGA data sets be merged for correlation analysis (Fig 5)? The melanoma cohorts (Gide and Liu) used in training can also be merged to identify possible more universal pathway-based biomarkers. This is especially relevant as there does not seem to be a substantial overlap among the top 10 most important features identified from the two datasets. The small effect size in the correlation (significant but rather weak, i.e., $|PCC| < 0.4$) also deserves a discussion. Does it imply existence of certain subtypes of the disease among the individuals of the cohort for instance (perhaps the meta-data from these cohorts can hint some insights)?

We thank the reviewer for this question. We used four different ICI-treated cancer cohorts (Gide, Liu, Kim, and IMvigor210) to train a NetBio-based machine-learning model. We then merged three TCGA cancer datasets (melanoma, gastric cancer, and bladder cancer) to compute the predicted ICI responses. We observed that correlations with immune cell proportions, especially CD8 T cells, were reduced when the merged TCGA dataset was used (Supplementary Fig.12, Supplementary Fig.13). Furthermore, when we used the gene expression levels of the network neighbors of ICI targets (PD1, PD-L1, and CTLA4) to measure transcriptome similarity between cohorts, we found that transcriptome similarity was high between two melanoma cohorts (median transcriptome similarity of 0.39 and 0.41 for

ICI responders and non-responders, respectively) and low between cohorts of different cancer types (Supplementary Fig.27). These results suggest that cancer type-specific immune cell infiltration patterns exist across cancer types. The new results have been added to the manuscript (page 14, line 8 and page 21, line 20).

Furthermore, although the predicted responders in two melanoma cohorts (Gide and Liu) showed high immune infiltration (Supplementary Fig.14), the correlation between predicted responses of patients in a TCGA melanoma cohort with the immune subtype based on models trained with the Gide and Liu cohorts, respectively, was weak (Supplementary Fig.15). Specifically, we trained a NetBio-based machine-learning model using the Gide or Liu cohort and then compared the predicted ICI responses with TCGA subtypes. We observed that predicted probability of ICI response was highest in patients with the immune subtype when either the Gide data (Supplementary Fig.14a) or the Liu data (Supplementary Fig.14b) were used, suggesting that predicted responders had high immune cell infiltration. Then, we asked whether ICI response predictions based on the Gide and Liu data were similar among melanoma patients with high immune infiltration. We found that even among melanoma patients with high immune infiltration, the ICI response predictions based on the Gide and Liu data were dissimilar (Supplementary Fig.15). These results suggest (i) that ICI responders may have distinct immune cell infiltration mechanisms and (ii) the existence of molecular subtypes within melanoma patients. These results are included in the manuscript on page 14, line 11.

Supplementary Figure 12. Correlation between NetBio-based predictions and immunogenomic features in the TCGA cohort. Correlation was measured using Pearson's correlation coefficient (PCC).

Supplementary Figure 13. Correlation between NetBio-based predictions and immunogenomic features in a merged TCGA cohort comprising melanoma (SKCM), gastric cancer (STAD), and bladder cancer (BLCA). Correlation was measured using Pearson's correlation coefficient.

Supplementary Figure 14. Association between predicted ICI response and TCGA subtypes in melanoma patients. (a)-(b) Associations measured using the (a) Gide or (b) Liu cohort to train a NetBio-based machine-learning model. Statistical significance was measured using Mann-Whitney U test. Boxplot shows median value, interquartile range (IQR) as bounds of the box and whiskers that extends from the box to upper/lower quartile \pm IQR \times 1.5.

Supplementary Figure 15. Correlation between predicted ICI responses of TCGA melanoma patients with the immune subtype based on different training datasets. Associations between ICI responses predicted using the Gide or Liu datasets to train a NetBio-based machine-learning model. Pearson correlation was used to statistical significance.

6. To show improvement over using only TME-Bio, the authors combine those biomarkers with Netbio and present the survival rates for the combination and TME-Bio. It would be good to see what the survival rates for Netbio only are to check whether including TME-Bio makes sense or not.

We thank the reviewer for this suggestion. We have included the performance of NetBio in predicting overall survival in the bladder cancer dataset (Supplementary Fig.21). We found that combining NetBio and TMB improved the predictive performance (Figure 7c) compared with using either NetBio (Supplementary Fig.21) or TMB (Figure 7b) separately, suggesting that both genomic and transcriptomic markers are useful in improving ICI response prediction.

Supplementary Figure 21. Prediction of overall survival among patients with available TMB levels in the IMvigor210 dataset. Predictive performance of Leave-One-Out Cross-Validation (LOOCV). Statistical significance was measured using log-rank test. Pred R, Predicted Responders. Pred NR, Predicted Non-Responders.

7. Some methodological clarification would help for the following: (i) “gene/pathway expression levels are z-score-standardized”, I have not been able to find details on how this was done, were the values for each gene/pathway standardized across the samples of the same cohort? (ii) what is the final value of w that was identified after optimization across the parameter space? (iii) what was the dumping factor in PageRank algorithm used in network propagation (currently the text states that default parameters of the package were used but

please specify as the implementation might change / be updated in the future).

We conducted z-score standardization of expression for each gene/pathway across the samples of the same cohort. We describe this method detail on page 28, line 22. The optimal hyperparameters identified during LOOCV are provided in the Source Data (page 28, line 17). We used 0.85 for the damping factor in the PageRank algorithm (page 26, line 19).

8. Overall, it is not clear what would be the overall Specificity, Sensitivity and Precision for a classifier that uses Netbio for ICIs across 3 different cancer types? Perhaps this can be included in the discussion. Furthermore, a supplementary table containing the list of genes in each of the Genebio, TME-Bio and Netbio would be certainly useful for further validation / exploitation of the study findings.

We have included a supplementary table that shows the specificity, sensitivity, and precision of our machine-learning model (Supplementary Table 2-5). We also provide the lists of genes for GeneBio, TME-Bio, and NetBio in the Source Data.

study	markers	accuracy	precision	F1	TP	TN	FP	FN	sensitivity	specificity
Gide	NetBio	0.73	0.74	0.75	37	29	13	12	0.76	0.69
Gide	PD1	0.67	0.70	0.69	33	28	14	16	0.67	0.67
Gide	PD-L1	0.71	0.74	0.73	35	30	12	14	0.71	0.71
Gide	CTLA4	0.66	0.70	0.67	32	28	14	17	0.65	0.67
Gide	GeneBio	0.71	0.74	0.73	35	30	12	14	0.71	0.71
Gide	CD8T	0.71	0.76	0.72	34	31	11	15	0.69	0.74
Gide	T exhaust	0.68	0.70	0.71	35	27	15	14	0.71	0.64
Gide	CAF	0.52	0.55	0.55	27	20	22	22	0.55	0.48
Gide	TAM	0.64	0.66	0.67	33	25	17	16	0.67	0.60
Gide	TME-Bio	0.65	0.67	0.68	34	25	17	15	0.69	0.60

Supplementary Table 2. Prediction performances of LOOCV using the Gide dataset to train a machine-learning model (l2 regularized logistic regression).

study	markers	accuracy	precision	F1	TP	TN	FP	FN	sensitivity	specificity
Liu	NetBio	0.64	0.54	0.58	30	46	26	17	0.64	0.64
Liu	PD1	0.07	0.03	0.03	2	6	66	45	0.04	0.08
Liu	PD-L1	0.42	0.30	0.32	16	34	38	31	0.34	0.47
Liu	CTLA4	0.55	0.45	0.50	26	40	32	21	0.55	0.56
Liu	GeneBio	0.52	0.40	0.40	19	43	29	28	0.40	0.60
Liu	CD8T	0.60	0.49	0.53	27	44	28	20	0.57	0.61
Liu	T exhaust	0.65	0.55	0.57	28	49	23	19	0.60	0.68
Liu	CAF	0.52	0.42	0.47	25	37	35	22	0.53	0.51
Liu	TAM	0.54	0.42	0.43	21	43	29	26	0.45	0.60
Liu	TME-Bio	0.55	0.43	0.46	23	42	30	24	0.49	0.58

Supplementary Table 3. Prediction performances of LOOCV using the Liu dataset to train a machine-learning model (l2 regularized logistic regression).

study	markers	accuracy	precision	F1	TP	TN	FP	FN	sensitivity	specificity
Kim	NetBio	0.84	0.78	0.67	7	31	2	5	0.58	0.94
Kim	PD1	0.51	0.27	0.35	6	17	16	6	0.50	0.52
Kim	PD-L1	0.31	0.28	0.44	12	2	31	0	1.00	0.06
Kim	CTLA4	0.44	0.31	0.47	11	9	24	1	0.92	0.27
Kim	GeneBio	0.40	0.27	0.40	9	9	24	3	0.75	0.27
Kim	CD8T	0.67	0.43	0.55	9	21	12	3	0.75	0.64
Kim	T exhaust	0.82	0.75	0.60	6	31	2	6	0.50	0.94
Kim	CAF	0.60	0.33	0.40	6	21	12	6	0.50	0.64
Kim	TAM	0.73	0.50	0.50	6	27	6	6	0.50	0.82
Kim	TME-Bio	0.82	0.70	0.64	7	30	3	5	0.58	0.91

Supplementary Table 4. Prediction performances of LOOCV using the Kim dataset to train a machine-learning model (l2 regularized logistic regression).

study	markers	accuracy	precision	F1	TP	TN	FP	FN	sensitivity	specificity
IMvigor210	NetBio	0.69	0.37	0.41	32	175	55	36	0.47	0.76
IMvigor210	PD1	0.64	0.30	0.36	30	160	70	38	0.44	0.70
IMvigor210	PD-L1	0.66	0.28	0.29	21	176	54	47	0.31	0.77
IMvigor210	CTLA4	0.62	0.28	0.33	28	157	73	40	0.41	0.68
IMvigor210	GeneBio	0.58	0.21	0.25	21	152	78	47	0.31	0.66
IMvigor210	CD8T	0.65	0.30	0.34	27	166	64	41	0.40	0.72
IMvigor210	T exhaust	0.57	0.25	0.31	29	141	89	39	0.43	0.61
IMvigor210	CAF	0.48	0.26	0.38	47	97	133	21	0.69	0.42
IMvigor210	TAM	0.64	0.31	0.39	34	156	74	34	0.50	0.68
IMvigor210	TME-Bio	0.60	0.27	0.33	29	150	80	39	0.43	0.65

Supplementary Table 5. Prediction performances of LOOCV using the IMvigor210 dataset to train a machine-learning model (l2 regularized logistic regression).

Minor:

Pg3:49 must be developed => is needed

We thank the reviewer for this comment. We updated the manuscript on page 3, line 9.

Pg3:57 many studies have reported => authors cite only 1 study

We have added additional references on page 3, line 18.

Pg15:313 $P < 10^{-16} \Rightarrow P < 10^{-15}$ (given that the lowest P was 1.5×10^{-16})

To avoid confusion in the statistical testing, we changed to pairwise tests between the three IHC phenotypes. (Figure 6b, c; page 16, line 21).

Fig. 6. The expression levels of NetBio pathways are consistent with immunohistochemistry-based immune phenotypes in bladder cancer. (a) Categorization of immune phenotypes using immunohistochemistry. **(b)–(c)** Expression levels of NetBio pathways in various immune phenotypes. For NetBio pathways, chemokine receptors bind chemokines **(b)** and FcγR activation **(c)** are shown. Two-sided Mann-Whitney U test was used to compute statistical significance for differential pathway expression levels across different immune phenotype patient groups. Boxplot shows median value, interquartile range (IQR) as bounds of the box and whiskers that extends from the box to upper/lower quartile $\pm IQR \times 1.5$.

Pg16:353-Pg17 raf activation \Rightarrow Raf activation (multiple occurrences)

We have updated the manuscript following the reviewer's suggestion.

Reviewer #4 (Remarks to the Author): Expert in immunotherapy, pathology, and artificial intelligence

In their manuscript „Network-based machine learning approach to predict immunotherapy

response in cancer patients“, Kong and colleagues describe a machine learning framework which they call NetBio and use this to identify novel predictive biomarkers for the response to immune checkpoint inhibitors. To use network biology is an interesting approach to address this important clinical challenge and the experiments are rather comprehensive. However, I have various major concerns:

Methodology

- In the first experiments (Figure 2) the authors compare their NetBio model with the expression levels of drug targets such as PD-1, PD-L1, and CTLA4. While I know that this is a heavily debated issue, in my opinion this is not sufficient as it stops at the transcriptional / translational level. Post-translational modifications, protein-protein interactions, intra- and extracellular localization, and most importantly tumor heterogeneity are all not considered. However, scoring systems, which are currently used in a clinical setting (such as TPS, CPS, and IC score) are based on immunohistochemistry (IHC) to determine exactly, which cells and tissues express these targets. It would be highly desirable to compare the NetBio model to these scores in the cohorts used.

We agree with the reviewer on comparing NetBio to IHC-based predictions. To compare IHC-based predictions with NetBio-based predictions, we used a cohort of Atezolizumab (PD-L1 inhibitor)-treated bladder cancer patients (PMID: 29443960) for which both bulk RNA sequencing data and tumor proportion score (TPS) are available. The proportions of PD-L1 expression on tumor cells, determined using IHC, were used to compute TPS. Across three different prediction tasks; leave-one-out cross-validation (LOOCV), Monte-Carlo cross validation (80% training, 20% testing for 100 independent iterations), and prediction of overall survival; we found that NetBio-based predictions performed better than TPS-based predictions (Supplementary Fig.4). In LOOCV predictions, we found that although predictive performance measured using accuracy was similar between NetBio (accuracy = 0.69) and TPS (accuracy = 0.67), there was an observable difference in F1 score (NetBio F1 = 0.41, TPS F1 = 0.24). This suggests that NetBio can better recover True Positives (patients who would benefit from Atezolizumab treatment), while also accurately predicting True Negatives (patients who would not benefit from Atezolizumab treatment). Similar results were observed when conducting Monte-Carlo cross validation for 100 different iterations, where NetBio-based predictions were significantly better compared with TPS-based predictions (Supplementary Fig.4b; P -value < 0.05 considered significant). Moreover, when using LOOCV to predict overall survival, whereas NetBio accurately predicted responders and non-responders (Supplementary Fig.4c; log-rank test P -value = 3.35×10^{-3}), TPS-based predictions were not a great predictor of overall survival (Supplementary Fig.4c; log rank test P -value = 0.37). While the utility of NetBio in clinical settings requires further investigation, our results suggest that NetBio can help improve prediction of ICI responses.

These results are now included in the manuscript (page 9, line 9).

Supplementary Figure 4. Comparing NetBio-based and tumor proportion score (TPS)-based predictions. (a) Predictive performance of Leave-One-Out Cross-Validation (LOOCV). (b) Predictive performance of Monte-Carlo cross-validation. Eighty percent of the samples were used as the training set, and the remaining 20% were used as the test set for 100 independent iterations. Statistical significance was measured using two-sided Student’s t-test. Boxplot shows median value, interquartile range (IQR) as bounds of the box and whiskers that extends from the box to upper/lower quartile $\pm IQR \times 1.5$. (c) Prediction of overall survival using either TPS or NetBio to train a machine-learning model (Logistic Regression). Statistical significance was measured using log-rank test. Pred R, Predicted Responders. Pred NR, Predicted Non-Responders.

- Furthermore, the authors claim, that they compared the NetBio model to two other network-based models called GeneBio and TME-Bio. However, throughout the manuscript, the comparisons are done for each “target” (PD1, PD-L1, CTLA4, CD8, T exhaust, etc.) individually. The authors should try training additional logistic regression models, which include all GeneBio and TME-Bio targets respectively. This would be a much “fairer” comparison.

We thank the reviewer for this excellent question. We made comparisons with logistic regression models that used (i) all of the drug targets (GeneBio) and (ii) all of the tumor microenvironment-associated biomarker genes (TME-Bio). Compared with NetBio-based models, the GeneBio-based and TME-Bio-based models were less predictive of patient responses to immunotherapy. Specifically, in Leave-One-Out Cross-Validation (LOOCV) spanning four different immunotherapy cohorts, GeneBio-based and TME-Bio-based predictions showed lower predictive performances compared with NetBio-based predictions (Figure 2). Using accuracy and F1 score to measure the predictive performances, we found that NetBio-based predictions were better in 71 of 72 comparisons (98.6%) with predictions using all other biomarkers, including the GeneBio and TME-Bio gene sets (Figure 2h-o).

Moreover, predictions using NetBio were on par with or better than GeneBio-based or TME-Bio-based predictions when fewer samples were used to train the machine-learning model (Supplementary Fig.3). For 100 iterations, when randomly selecting 80% of the samples for training and the remaining 20% for testing, we found that NetBio-based predictions had significantly better or equal performance in 70 of 72 comparisons (97.2%).

Also, we confirmed that GeneBio-based and TME-Bio-based models were less predictive of ICI response in across-study predictions. When we used the Gide dataset to train the machine-learning model and tested the predictive performance on three independent cohorts (Auslander, Prat, and Riaz), GeneBio-based predictions (average AUC = 0.68, average AUPRC = 0.47; Figure 3, Supplementary Fig.5) and TME-Bio-based predictions (average AUC = 0.62, average AUPRC = 0.33; Figure 3, Supplementary Fig.5) performed less optimally compared with NetBio-based predictions (average AUC = 0.74, average AUPRC = 0.63; Figure 3, Supplementary Fig.5). Furthermore, when we trained the machine-learning model for 100 different iterations using 80% of the randomly sampled patients from the training dataset (Gide dataset), NetBio-based predictions showed significantly better or equal performance in 23 of 27 comparisons (85.2%) using AUC as a performance metric (Supplementary Fig.9b) and 26 of 27 comparisons (96.3%) using AUPRC as a performance metric (Supplementary Fig.9c). These results suggest that NetBio-based machine-learning can make robust predictions compared with machine learning based on expression levels of other ICI-response-associated biomarkers.

These results are now shown in Figure 2, Figure 3, Supplementary Fig.5, and Supplementary Fig.9.

- Additionally, there are more “sophisticated” models than l2-regularized logistic regression. The authors should consider trying out other methods such as SVMs, random forests, or even neural network based approaches.

We thank the reviewer for this great comment. As the reviewer suggested, we compared the predictive performances of support vector machine classifier (SVC), random forest (RF), and deep neural network (DNN) models with that of the logistic regression (LR)

model. We found that (i) the predictive performance of machine-learning (ML) models trained using SVC or RF was similar to that of the LR-based model and (ii) the LR-based model was more generalizable to new datasets than the DNN model. First, we tested the predictive performance by conducting within-study cross validation (leave-one-out cross validation) and observed statistically identical performances among all four models (Supplementary Fig.33). All of the models used gene expression levels of NetBio features, and predictive performances using accuracy and F1 score across four different cohorts were used as performance metrics. The results showed that there were no statistically significant differences in performance when the ML models were used to predict ICI responses of patients from the same cohort (Supplementary Fig.33b; P-value < 0.05 considered significant). By contrast, for across-study predictions, we found that although the SVC-based and RF-based models performed similarly to the LR-based model, the LR-based model outperformed the DNN-based model (Supplementary Fig.34). Specifically, we used the Gide cohort to train the models and four different melanoma cohorts to test the LR-based and DNN-based models, and we identified significant differences between the performances of the two models (Supplementary Fig.34b; $P = 4.9 \times 10^{-4}$). These results suggest that DNN-based models may overfit to the training cohort, leading to reduced generalizability. While improving the use of DNN-based ML models to predict ICI response is an important topic for future research, it is possible that more ICI data may be required to use DNN-based ML models.

These results are now included in the manuscript (page 28, line 7).

Supplementary Figure 33. Comparison of LOOCV performance using NetBio machine-learning models based on Logistic Regression (LR), support vector machine classifier (SVC), random forest (RF), or deep neural network (DNN). (a) Overall scheme of immunotherapy response prediction using LOOCV. (b) Summarized classification results. Statistical significance was measured using paired Student's t-test. Boxplot shows median value, interquartile range (IQR) as bounds of the box and whiskers that extends from the box to upper/lower quartile $\pm IQR \times 1.5$. (c)-(j) LOOCV predictive performance based on (c)-(f) accuracy and (g)-(j) F1 score.

Supplementary Figure 34. Comparison of across-study predictive performance using NetBio machine-learning models based on Logistic Regression (LR), support vector machine classifier (SVC), random forest (RF), or deep neural network (DNN). (a) Overall scheme of across-study immunotherapy response prediction. (b) Summarized classification results. Statistical significance was measured using paired Student's t-test. Boxplot shows median value, interquartile range (IQR) as bounds of the box and whiskers that extends from the box to upper/lower quartile $\pm IQR \times 1.5$. (c)-(j) Across-study predictive performances using (c)-(f) AUC and (g)-(j) AUPRC.

Performance

- While there seems to be an improvement of the predictive performance of the NetBio model when compared to other transcriptomic features (PD1, PD-L1, etc.), the achieved values are generally rather low. This is even more worrying, as the number of cases is somewhat small (see I. e. ROC curve in Figure 3 b) and the class distribution is not directly obvious. The AUROC can overestimate the performance of a model, especially if the number of responders

/ non-responders is not balanced. Precision-Recall-Curves (with the indication of the class weight) and cross tables could help to better understand the performance of each approach. Consequently, the ranks displayed in Figure 4 are not really meaningful to me, as a) the comparison is biased towards the NetBio approach (see comment above) and b) the performance even of the best ranking model is not really high.

We thank the reviewer for this comment. We agree with the reviewer that the predictive performance can be improved. We now show that NetBio outperforms other methods (i) when using area under the precision-recall curve (AUPRC) as a performance metric and (ii) when other state-of-the-art methods are used. First, we showed that NetBio-based predictions outperformed drug-target-based or tumor microenvironment-based predictions when AUPRC was used as a prediction performance metric (Supplementary Fig.5). We have added distributions of prediction results (Supplementary Fig.35) as well as cross tables (Supplementary Tables 2-5). Moreover, in order to further demonstrate the improvement of our method over other approaches, we compared our NetBio predictions with those of other state-of-the-art immunotherapy response prediction methods (PMID: 30127393, PMID: 34430923, PMID: 34019806, PMID: 30127394) and a deep neural network-based method (PMID: 31825821). When using accuracy and F1 score as metrics of performance for LOOCV (Supplementary Fig.10), we found that NetBio-based predictions were better in 33 of 34 comparisons (97.1%). Also, we observed that NetBio-based predictions were better in 17 of 18 comparisons (94.4%) of across-study predictions (Supplementary Fig.11). These results suggest that (i) performance comparisons are not biased towards NetBio and (ii) NetBio can identify robust biomarkers that improve the prediction of ICI treatment response.

Additionally, following the reviewer's comment, we have removed the rank comparisons from the manuscript.

Supplementary Figure 5. Predictive performance in three independent melanoma datasets. (a) Overall scheme of immunotherapy response prediction in three independent datasets. The datasets used to train and test the machine-learning models, transcriptomic features used to train the models, and number of samples for each dataset are displayed. (b)-(d) The area under the precision-recall curve (AUPRC) for the (b) Auslander, (c) Prat, and (d) Riaz datasets are shown. No expression profiles of cancer-associated fibroblast (CAF) marker genes were available in the Prat dataset. N.D., Not detected.

Supplementary Figure 10. Comparison of Leave-One-Out Cross-Validation (LOOCV) performances. (a) Overall scheme of immunotherapy response prediction using LOOCV. (b)–(c) Summarized classification results using (b) accuracy and (c) F1 score as a metric to quantify predictive performance. (d)–(k) LOOCV predictive performances based on (d)–(g) accuracy and (h)–(k) F1 score.

Supplementary Figure 11. Comparison of across-study predictive performance. (a) Overall scheme of across-study predictions of immunotherapy response. (b)–(c) Summarized prediction results using (b) area under the receiver operating characteristic curve (AUC) and (c) area under the precision-recall curve (AUPRC) as a metric to quantify predictive performance. (d) –(k) Across-study predictive performance based on (d)–(g) AUC and (h)–(k) AUPRC.

Novelty

- In Figure 5 the authors show that there is a strong overlap between known predictors of ICI response and the NetBio model. This is unsurprising to me as a) TMB is such a basic and general parameter and b) ICI responses are so heavily dependent on various immune cell populations. The connection to FGFR signaling is interesting but not further explored.

Following the reviewer's suggestion, we further explored correlations between expression levels of the FGFR signaling pathway and molecular phenotypes in melanoma patients. In agreement with our previous finding (Figure 5d), we discovered that melanoma patients with high immune infiltration ("Immune" subtype) exhibited lower FGFR pathway expression compared with patients with other melanoma subtypes (Supplementary Fig.18). Our result further suggests that FGFR signaling is associated with immune cell infiltration in melanoma patients.

These results are now included in the manuscript (page 15, line 6).

Supplementary Figure 18. Expression level of 'signaling by FGFR' pathway among TCGA melanoma subtypes. Statistical significance was measured using Mann-Whitney U test. Boxplot shows median value, interquartile range (IQR) as bounds of the box and whiskers that extends from the box to upper/lower quartile $\pm IQR \times 1.5$.

Clinical applicability

- To me the most important aspect is how the NetBio model compares to or could be included in existing ICI prediction algorithms used in a clinical setting. As discussed above, most of these are based on IHC to account for tumor heterogeneity and morphology. In Figure 6 the authors claim consistency with IHC-based immune phenotype in bladder-cancer. However, it remains completely unclear how this was determined (no mention of IHC in the methods section of the paper at all). Examples of the IHC stainings used would be highly desirable. Furthermore, the box plots in Figure 6 b and c are curious. Where does the p-value of 8.21×10^{-17} (!) come from? The error bars show considerable overlapping for a value this small. During ANOVA, did the authors account for a) the values distribution? b) the values standard deviation? c) multiple comparisons?

We thank the reviewer for this comment. We have updated the manuscript to contain the detailed method used for the analysis of IHC-based immune phenotypes in bladder cancer. The updated paragraphs are included in the Methods section of the manuscript (page 30, line 21).

Furthermore, in order to clarify the ANOVA results, we have added pairwise comparisons between IHC-based immune phenotypes in bladder cancer. Specifically, we conducted pairwise comparisons of the expression levels of NetBio pathways between

immune phenotypes. We found that the expression levels of the NetBio pathways were correlated with CD8 T cell infiltration levels (Figure 6b-c). Specifically, expression of the NetBio pathways was higher in the immune infiltrated subtype than in the immune desert or excluded subtypes, suggesting that NetBio pathways can capture relevant information about immune cell infiltration.

Fig. 6. The expression levels of NetBio pathways are consistent with immunohistochemistry-based immune phenotypes in bladder cancer. (a) Categorization of immune phenotypes using immunohistochemistry. **(b)–(c)** Expression levels of NetBio pathways in various immune phenotypes. For NetBio pathways, chemokine receptors bind chemokines **(b)** and FcγR activation **(c)** are shown. Two-sided Mann-Whitney U test was used to compute statistical significance for differential pathway expression levels across different immune phenotype patient groups. Boxplot shows median value, interquartile range (IQR) as bounds of the box and whiskers that extends from the box to upper/lower quartile \pm IQR \times 1.5.

- The data on the addition of the NetBio model together with TMB in Figure 7 is interesting. However, while there was an initial hype in using TMB as a biomarker for ICI response, the importance of this parameter in a clinical setting has dramatically decreased recently. So, the performance increase would have to be really pronounced to include the use of both outside of a research setting.

We agree with the reviewer that TMB is an important, yet insufficient, predictor of ICI response. Indeed, we observed that a model combining TMB and NetBio performed better than a model using only TMB to predict overall survival (Supplementary Fig.22). For example, we found that ICI-non-responders with high TMB could be correctly classified using the combined TMB and NetBio model (Fig S22a,b and Fig S23). Specifically, we found that the combined model captured a subgroup of patients with high TMB who displayed lower overall survival (Fig S22b; log-rank test $P = 0.07$), suggesting that non-responders exist among patients with high TMB. Our finding is in line with previous reports that not all

patients with high TMB respond to ICI treatment (PMID: 33199494, PMID: 32320754). Also, we were able to detect ICI responders among patients with low TMB (Fig S22a,c and Fig S23) when using the combined model. Specifically, a subgroup of low-TMB patients with better overall survival was identified using the combined model (Fig S22c; $P = 1.94 \times 10^{-2}$). This result suggests that patients who would benefit from ICI treatment despite having low TMB can be better identified when both TMB and NetBio are used for prediction. Altogether, our results suggest that combining NetBio and TMB can improve accuracy in the classification of responders and non-responders.

Supplementary Figure 22. Comparison of predictions from TMB-based PD-L1-inhibitor response prediction and TMB plus NetBio-based prediction. (a) Drug response class changes between TMB-based predictions and TMB- and NetBio-based predictions. **(b)–(c)** Overall survival differences between TMB-based predictions and TMB- and NetBio-based predictions. Statistical significances were measured using the log-rank test.

Supplementary Figure 23. Distribution of TMB levels among predicted ICI responders

and non-responders in the IMvigor210 dataset. The Mann-Whitney U test was used to measure statistical significance. Boxplot shows median value, interquartile range (IQR) as bounds of the box and whiskers that extends from the box to upper/lower quartile $\pm IQR \times 1.5$.

- The determination of the ground truth, meaning the identification of responders and non-responders is a key aspect of this study and can pose a serious challenge. The authors write, that RECIST criteria were used in some of the patients. How many? Then the authors state: “For datasets that did not provide or use RECIST criteria, we used responder and non-responder classification from the datasets.”. What exactly is meant by this statement? For how many patients was this “method” used to determine the ground truth (response vs. non-response)?

RECIST criteria-based responses were used for the Liu, Gide, IMvigor210, Kim, Riaz, and Prat datasets. We have included a supplementary table to clarify the drug response labels used throughout the manuscript (Supplementary Table 1). For the Auslander dataset, we used the responder and non-responder labels provided in the original manuscript (PMID: 30127394). We have included further information on the labeling of responders and non-responders on page 24, line 18.

Cohort name	Cancer type	Drug	Drug response label	Within or across study analysis
Liu	melanoma	Nivolumab, Pembrolizumab	RECIST	within study, across study
Gide	melanoma	Pembrolizumab, Nivolumab, Ipilimumab	RECIST	within study, across study
IMvigor210	bladder cancer	Atezolizumab	RECIST	within study
Kim	gastric cancer	Pembrolizumab	RECIST	within study
Auslander	melanoma	anti-PD-1, anti-CTLA-4	responder / non-responder	across study
Riaz	melanoma	Nivolumab	RECIST	across study
Prat	melanoma	Nivolumab, Pembrolizumab	RECIST	across study
Huang	melanoma	Pembrolizumab	recurrence	across study

Supplementary Table 1. Cohorts and drug response labels used in this study. For cohorts that used RECIST criteria, we considered patients with Complete Response (CR) or Partial Response (PR) as responders and those with Stable Disease (SD) or Progressive Disease (PD) as non-responders.

Other aspects

- There is no section that summarizes the statistical methods used in this manuscript. There is no statement on whether the underlying code can be accessed (i.e., within a GitHub repo, etc.). Furthermore, there is no statement on any ethics approval or that no ethics approval was necessary.

We have summarized the statistical methods in the Methods section (page 32, line 17). Also, a link to a GitHub repository that provides the source codes for reproduction of the results is now provided (page 33, line 15). Also, because we did not generate any new data, no ethics approval was necessary. This statement has been added to the manuscript (page 24, line 20).

REVIEWERS' COMMENTS

Reviewer #1 (Remarks to the Author):

None.

Reviewer #2 (Remarks to the Author):

The authors have resubmitted a manuscript that is very responsive to reviewer feedback, including new analysis and clarification/refinement of the analytical approach.

The melanoma datasets utilized make more sense and the testing/validation schemes flow more logically.

Fig 2 focuses on NetBio vs PD1 or CTLA4 given assessment of upstream genes for therapeutic target and outperforms these for the most part. Accuracy (F1 score) is then compared for NetBio vs other approaches in each dataset. The Y axes values change for each analysis. While NetBio outperforms the other approaches, for the most part this is not dramatic and still resides in the 0.4 – 0.7 range. I'm not sure this "outperformance" will lead to a clinically actionable assay or approach.

Fig 3 compares NetBio in other datasets, outperforming (barely) PD1, but the AUC is still in the 0.69 – 0.79 range. Again, it is not clear to me that this outperforms existing approaches by a wide margin.

Fig 5, they do correlate their findings with TCGA classes (lymphocyte high), which is reassuring.

The transition to Bladder is more smooth in this case, but also still remains a bit of a disconnect for me clinically. They do show improvement over TMB by addition of NetBio.

It is interesting that NetBio does not perform well in the setting of treatment (post-treatment tumors). Most previous work has demonstrated that on-treatment signatures are far more demonstrative of response than the untreated (pre-treatment tumors) since the biology is begging to declare itself in the challenged state.

https://ascopubs.org/doi/abs/10.1200/JCO.2017.35.15_suppl.9579

I will defer to the technical reviewers for methodological consideration. However, my clinical concern rests with what value this approach adds over existing and emerging methods. They do seem to define an immune high tumor subset that may be more predisposed to ICI response, but the performance and consistency is not enough to change the potential clinical utility of this in guiding treatment decision-making.

<https://pubmed.ncbi.nlm.nih.gov/35243413/>

However, the revised manuscript is significantly improved over initial submission, both in content and presentation.

Reviewer #3 (Remarks to the Author):

I thank the authors for addressing all of my previous comments in detail. The additional analyzes included in the manuscript help providing a more comprehensive understanding of the value of network-based biomarkers compared to conventional gene-based & microenvironment based ICI biomarkers as well as the current upper-bounds to their prediction capacity.

A relatively minor point that remains for exploration is the effect of the network propagation algorithm employed on this prediction capacity. The methods based on random walk with restart, such as page-rank used in the study is known to be prone to degree bias (nodes highly connected to each other in the protein interaction network) as demonstrated previously in the literature (Cowen et al, 2017, Nat Rev Gen, doi:10.1038/nrg.2017.38). Accordingly, it would be interesting to see whether using network propagation strategies accounting for degree bias such as DADA and GUILD (dois: 10.1186/1756-0381-4-19 ; 10.1371/journal.pone.0043557) has a significant impact on the identified biomarkers & associated survival prediction accuracy.

Reviewed by Emre Guney.

Reviewer #4 (Remarks to the Author):

The authors should be commended for their effort and all the additional work they put into the manuscript. They have addressed my concerns as far as this was possible. With the additional data, I would now be in favor of accepting this manuscript for publication, if the other reviewers and editors agree.

Reviewer #2 (Remarks to the Author):

The authors have resubmitted a manuscript that is very responsive to reviewer feedback, including new analysis and clarification/refinement of the analytical approach.

The melanoma datasets utilized make more sense and the testing/validation schemes flow more logically.

Fig 2 focuses on NetBio vs PD1 or CTLA4 given assessment of upstream genes for therapeutic target and outperforms these for the most part. Accuracy (F1 score) is then compared for NetBio vs other approaches in each dataset. The Y axes values change for each analysis. While NetBio outperforms the other approaches, for the most part this is not dramatic and still resides in the 0.4 – 0.7 range. I'm not sure this "outperformance" will lead to a clinically actionable assay or approach.

Fig 3 compares NetBio in other datasets, outperforming (barely) PD1, but the AUC is still in the 0.69 – 0.79 range. Again, it is not clear to me that this outperforms existing approaches by a wide margin.

Fig 5, they do correlate their findings with TCGA classes (lymphocyte high), which is reassuring.

The transition to Bladder is more smooth in this case, but also still remains a bit of a disconnect for me clinically. They do show improvement over TMB by addition of NetBio.

It is interesting that NetBio does not perform well in the setting of treatment (post-treatment tumors). Most previous work has demonstrated that on-treatment signatures are far more demonstrative of response than the untreated (pre-treatment tumors) since the biology is begging to declare itself in the challenged state.

https://ascopubs.org/doi/abs/10.1200/JCO.2017.35.15_suppl.9579

I will defer to the technical reviewers for methodological consideration. However, my clinical concern rests with what value this approach adds over existing and emerging methods. They do seem to define an immune high tumor subset that may be more predisposed to ICI response, but the performance and consistency is not enough to change the potential clinical utility of this in guiding treatment decision-making.

<https://pubmed.ncbi.nlm.nih.gov/35243413/>

However, the revised manuscript is significantly improved over initial submission, both in content and presentation.

The authors appreciate the reviewer's comment and agree that improved prediction in cooperation with other omics layers will help the better clinical decision immunotherapy responses, as demonstrated by PMID: 35243413. As the reviewer elaborated, we are aware of the fact that a strategy based on a single omics data has a limitation in explaining and interpreting biological complexity of clinical outcomes. Therefore, we tried to combine NetBio-based predictions with TMB or gene-gene synthetic lethal interactions to improve prediction performances of patients' response to immunotherapy. (**Fig. 7, Supplementary Figure 30**). These results suggest that NetBio-based predictions have a potential to advance guideline of treatment decision-making with improved predictive performance when NetBio is integrated with other methods indicate NetBio can capture responders / non-responders who are not detected by other features. In addition, advances and applications of various machine learnings on biological data bring upon a new opportunity for biomarker discovery, diagnosis, and patient stratification for treatment decision-making. [PMID: 33508232, PMID: 34019806, PMID: 33204028] We anticipate that development of network-based strategies together with diverse features from various omics layers will achieve more accurate diagnosis with the clinical benefit for immunotherapy patients prior to treatments.

We now add a discussion of this matter to the manuscript.

“Since biological outcomes of immunotherapy are highly complex, a method relying on a single omics feature has a limitation in predicting patient response to immunotherapy treatments. Combining a network-based machine learning model with diverse omics layers would make better clinical results.”

We also agree with the reviewer that on-treatment or post-treatment signatures would be better demonstrative for characterizing the responses. Of note, throughout the manuscript, NetBio-based prediction was built upon pre-treatment data since our aim was not to characterize but to predict the responses, which contribute to clinical decision whether the ICI treatment would better be initiated. Regarding on-treatment signatures, for the sake of comparison to NetBio, we extracted DEPs that relied on standard statistical procedure, lacking sophisticated feature selection like the network-based refinement for the relevance to ICI drugs (**Supplementary Fig.32**). Expanding network-based approach for on-treatment data remains a matter of

scientific exploration, since characterization of the ICI responses could be another important aim, which will be helpful for clinical decision-making to continue the treatment.

Reviewer #3 (Remarks to the Author):

I thank the authors for addressing all of my previous comments in detail. The additional analyzes included in the manuscript help providing a more comprehensive understanding of the value of network-based biomarkers compared to conventional gene-based & microenvironment based ICI biomarkers as well as the current upper-bounds to their prediction capacity.

A relatively minor point that remains for exploration is the effect of the network propagation algorithm employed on this prediction capacity. The methods based on random walk with restart, such as page-rank used in the study is known to be prone to degree bias (nodes highly connected to each other in the protein interaction network) as demonstrated previously in the literature (Cowen et al, 2017, Nat Rev Gen, doi:10.1038/nrg.2017.38). Accordingly, it would be interesting to see whether using network propagation strategies accounting for degree bias such as DADA and GUILD (dois: 10.1186/1756-0381-4-19 ; 10.1371/journal.pone.0043557) has a significant impact on the identified biomarkers & associated survival prediction accuracy.

Reviewed by Emre Guney.

We agree with the reviewer that the choice of propagation algorithms would affect the prediction performance. As the reviewer mentioned, we are aware of the fact that various network propagation algorithms have been tested on diverse tasks of precision medicine. We anticipate that the prediction of immunotherapy response is helped by continuous development of network propagation methods.

We now add a discussion of this matter to the manuscript.

“Additionally, continuous development of network propagation algorithms will help improve tasks of precision medicine since the algorithms have been successfully applied to identify disease genes and drug targets [PMID: 28607512]. In this study, a random walk with restart was employed. However, various algorithms of network propagation have been recently proposed to account for degree bias of protein interaction networks. [PMID: 21699738, PMID: 23028459]. These methods have a potential to find diseases modules with improved performance of identifying disease genes, drug target candidates, and biomarkers for drug response.”

We hope that the reviewer kindly understands that as we go forward in this type of research, we are keenly aware of the issues and research opportunities in dealing with developments in network biology research. Furthermore, to the best of our knowledge, we have made a valuable contribution.